# Ocean colour signature of climate change

Stephanie Dutkiewicz [1,2], Anna E. Hickman[3], Oliver Jahn[1], Stephanie Henson[4], Claudie Beaulieu[3,5] & Erwan Monier [2,6]

Monitoring changes in marine phytoplankton is important as they form the foundation of the marine food web and are crucial in the carbon cycle. Often Chlorophyll-a (Chl-a) is used to track changes in phytoplankton, since there are global, regular satellite-derived estimates. However, satellite sensors do not measure Chl-a directly. Instead, Chl-a is estimated from remote sensing reflectance ($R_{RS}$): the ratio of upwelling radiance to the downwelling irradiance at the ocean's surface. Using a model, we show that $R_{RS}$ in the blue-green spectrum is likely to have a stronger and earlier climate-change-driven signal than Chl-a. This is because $R_{RS}$ has lower natural variability and integrates not only changes to in-water Chl-a, but also alterations in other optically important constituents. Phytoplankton community structure, which strongly affects ocean optics, is likely to show one of the clearest and most rapid signatures of changes to the base of the marine ecosystem.

[1] Department of Earth, Atmospheric and Planetary Sciences, Massachusetts Institute of Technology, Cambridge, MA 02139, USA. [2] Center for Climate Change Science, Massachusetts Institute of Technology, Cambridge, MA 02139, USA. [3] Ocean and Earth Sciences, University of Southampton, National Oceanography Centre Southampton, Southampton SO14 3ZH, UK. [4] National Oceanography Centre Southampton, Southampton SO14 3ZH, UK. [5] Ocean Sciences Department, University of California, Santa Cruz, CA 95064, USA. [6] Present address: Department of Land, Air and Water Resources, University of California, Davis, CA 95616, USA. Correspondence and requests for materials should be addressed to S.D. (email: stephd@mit.edu)

Phytoplankton in the sunlit layer of the ocean are important both as the base of the marine food web, and so fuelling fisheries, and in regulating key biogeochemical processes such as export of carbon to the deep ocean. Satellite ocean colour measurements over the last two decades have allowed the scientific community an unprecedented dataset to study phytoplankton on a global scale and at regular, frequent intervals. Ocean colour satellite sensors measure the radiance at the top of atmosphere over a range of wavelengths. After taking account of the optically significant constituents in the atmosphere (which can include a substantial error)[1], a key product of ocean colour is remotely sensed reflectance ($R_{RS}$), the ratio of the upwelling radiance to the downwelling irradiance at the ocean surface. $R_{RS}$ is the standard product provided by space agencies. Several algorithms and quasi-analytical methods are used to deduce more ecologically relevant quantities (such as Chl-a, the main pigment utilized in photosynthesis) from the $R_{RS}$ measurements[2–4]. Such ocean colour products have been used to explore trends in ocean surface Chl-a[5–7] as well as primary production[8], suggesting complex, but as yet limited, patterns of long-term change.

Numerical models provide a means to explore potential future changes in phytoplankton due to anthropogenic climate change. These models[9–14] in general suggest a decrease in globally integrated primary productivity driven by a reduction in supply of macronutrients, though the predicted changes vary in magnitude[12,13], and some regions have increased productivity.

However, a key question remains: How long will it take to detect an unambiguous signal of climate change in phytoplankton populations? Modelling studies have been used to caution that it will take many decades for significant trends in Chl-a and primary production to be detectable[15,16]. This is due to the magnitude of the signal of change relative to the often-large interannual-to-decadal variability in these quantities. Thus, while satellite ocean colour products provide regular and global coverage, even these data will require additional decades of observations before the signal of climate change is obvious over large regions of the ocean.

Moreover, the satellite ocean colour products of Chl-a and primary production are still proxies (based on $R_{RS}$ measurements) of the real quantities (as might be measured in situ). How are other optically important water constituents predicted to alter over the coming century and how do they together alter reflectance and ocean colour, as observed by satellites? Put another way: How does the colour of the ocean change?

Here, we use a unique ocean physics, biogeochemistry and ecosystem model that explicitly includes a representation of the ocean's optical properties[17] to explore how climate change is manifested in ocean colour over the course of the 21st century. Because of the inclusion of a radiative transfer component, the model captures how light penetrates through and is scattered back out of the ocean, and can therefore calculate $R_{RS}$ (ocean colour). In this paper, we specifically address how strong the climate change signal will be in ocean colour and the different factors that affect it. We determine which optical property is likely to respond most rapidly to climate change, and thus should be the focus of efforts to detect robust climate-driven trends in satellite ocean colour records. The model also allows for estimating Chl-a from the model $R_{RS}$ in a similar way to the typical real world ocean colour Chl-a product[2]. As such, we use the model to consider the implications of using satellite-derived Chl-a for monitoring climate trends.

## Results

### The present day and interannual variability.
The current day biogeochemical and ecosystem model fields have been validated against and compare well to observations (see Methods, Supplementary Figs. 1–4; also see previous papers using this model[13,14,17–19]). Chl-a (Fig. 1a) has high values in subpolar regions and along the equator where upwelling water supplies nutrients to fuel the marine ecosystem as is found in the real ocean (Fig. 1c). These regions are dominated by larger phytoplankton cells (Supplementary Fig. 5a) such as diatoms. Chl-a is much lower in the subtropical gyres; here nutrient supplies are low and pico-phytoplankton, with their high nutrient affinity, dominate.

A unique feature of this model is the explicit parameterization of upwelling and downwelling irradiance, such that we can calculate $R_{RS}$ (see Methods). In the model, $R_{RS}$ is resolved over the visible spectrum from 400 to 700 nm in 25 nm bands. We note that the model $R_{RS}$ does not have the uncertainties that the real world $R_{RS}$ has due to the atmospheric correction[1]. The model does not have the exact same wavebands as the ocean colour satellites, therefore we interpolate the model $R_{RS}$ to the same bands as the satellite measurements (Fig. 2, Supplementary Fig. 2). The model captures the reversed patterns between blue (443 nm) and green (555 nm) $R_{RS}$ between gyres and highly productive regions. The model underestimates the blue $R_{RS}$ in the subtropics where modelled Chl-a is likely too low relative to the real ocean (Fig. 1, Supplementary Fig. 2), and the effects of salinity on the ocean optics, not resolved in the model, may become more important[20]. The model has noticeably higher green (550 nm) $R_{RS}$ in the equatorial Atlantic and Indian Ocean than the satellite measurements, but this is consistent with the model over-estimating Chl-a relative to the satellite product in this region (Fig. 1, Supplementary Fig. 2). These are also regions of high cloud cover where the satellite product may be biased.

We construct a model ocean colour Chl-a product, similar to that provided from an often used ocean colour algorithm[2]. This proxy for Chl-a is derived from the model reflectance fields and will be called derived Chl-a. This is a different property to model actual Chl-a which is the sum of the dynamic Chl-a that is explicitly resolved and is thus more equivalent to the Chl-a that would be measured in situ. The derived Chl-a links the model blue/green reflectance ratio to model actual Chl-a in a manner equivalent to the algorithm that is often used in ocean colour products in the real world[2,19] (see Methods). This blue/green ratio algorithm has co-variations with Chl-a, CDOM and detrital matter intrinsically built into it[19,21,22]. This product (Fig. 1b) is more appropriate to evaluate against real world satellite-derived Chl-a (Fig. 1c, Supplementary Fig. 2) as it is a more equivalent property[19] and it captures the features of the model actual Chl-a well.

Ocean ecosystems, and therefore ocean colour, are not static, changing with the seasons and interannually (Fig. 1d–f, Supplementary Fig. 3). The model (Fig. 1e) in general captures the patterns of the satellite estimated variability (Fig. 1f), though overestimates the interannual variability in higher latitudes, where it also overestimates the mean (Supplementary Fig. 2). The model Equatorial Pacific has a narrower band of variability around the equator than seen in the observations (Supplementary Fig. 3), a discrepancy that shows up in both Chl-a and $R_{RS}$. The $R_{RS}$ variability is otherwise generally underestimated. We find it instructive to also examine the magnitude of the interannual variability relative to the climatological mean composite (Fig. 3, Supplementary Fig. 4). The model's slightly lower value relative to the observed median magnitude of this ratio in Chl-a (Fig. 3a) is expected as the model does not capture all the sources of variability (e.g. mesoscale features) found in the real ocean. The current day relative magnitude of interannual variability of the other optically important constituents (CDOM, detrital particles) are predicted by the model to have similar values to Chl-a.

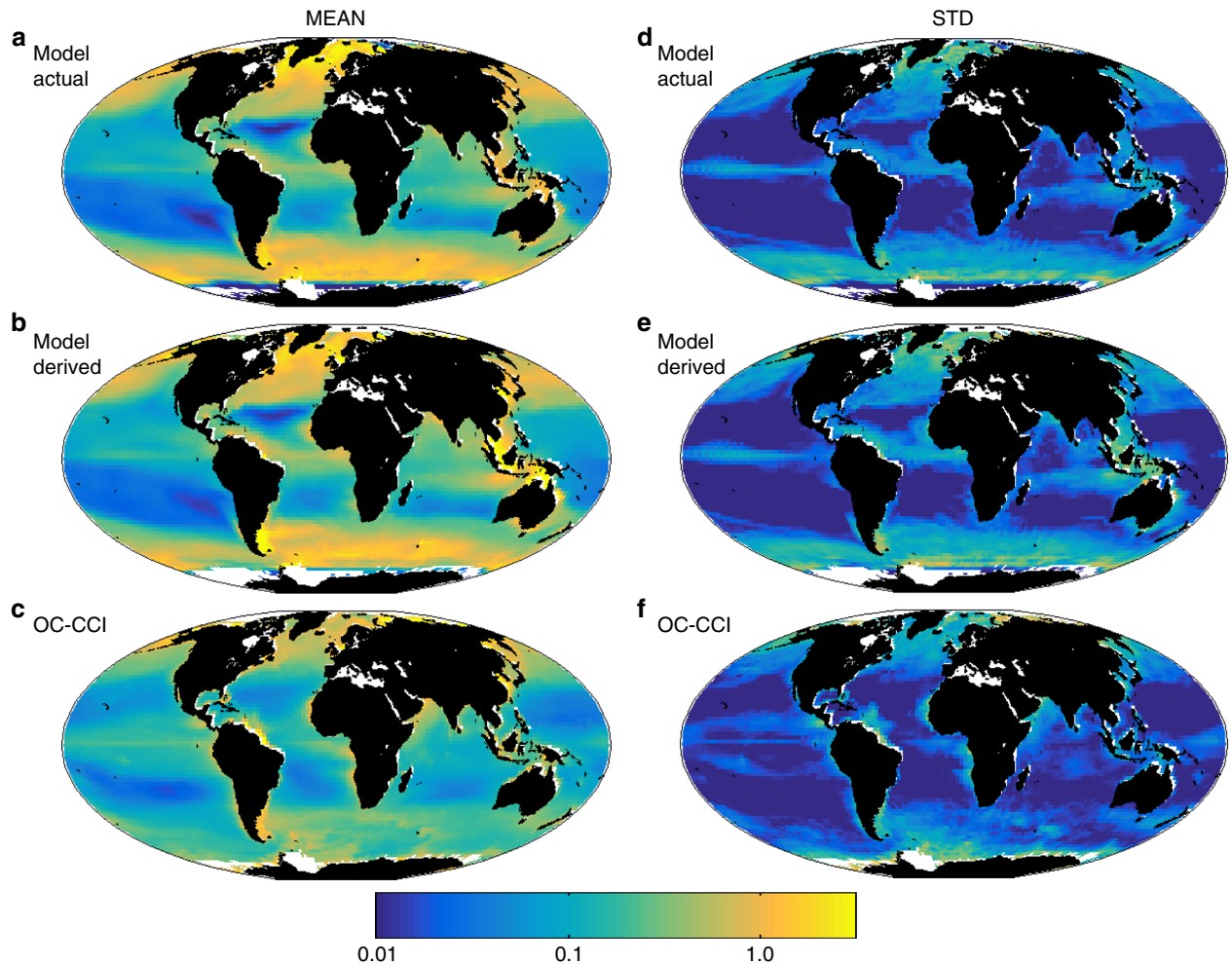

**Fig. 1** Current day Chl-a and its interannual variability. Composite mean Chl-a (mg Chl m$^{-3}$) for 1998–2015: **a** model actual; **b** model satellite-like derived (using an algorithm and the model $R_{RS}$); **c** Ocean Colour Climate Change Initiative project (OC-CCI, v2) satellite derived. Interannual variability defined as the standard deviation of the annual mean composites (1998–2015): **d** model actual; **e** model satellite-like derived; **f** OC-CCI, v2 satellite derived. White areas are regions where model resolution is too coarse to capture the smaller seas, or where there is persistent ice cover. Statistical comparison of derived model and OC-CCI product are provided in Supplementary Figs. 1–3. Model actual Chl-a is the sum of the dynamic Chl-a for each phytoplankton type that is explicitly resolved in the model. It is equivalent to the Chl-a that would be measured in situ. This is distinct to satellite-derived Chl-a which is calculated via an algorithm derived from the reflected light measured by ocean colour satellite instruments

However, the relative magnitude of the observed interannual variability of $R_{RS}$ is lower (red symbols in Fig. 3b) than these other variables. The model captures these lower ratios (though underestimates the variability, as in Chl-a) over most of the spectrum, but not at the high wavelength bands (Fig. 3b, Supplementary Fig. 4). However, we note that the uncertainty in $R_{RS}$ is higher at higher wavelength bands[23].

**Physical, biogeochemical and ecosystem response**. The goal of this paper is not to repeat a detailed discussion of the modelled biogeochemical and ecological responses to climate change, however we briefly summarize the main features here. In the business as usual scenario (similar to the IPCC RCP8.5[24]), the model marine ecosystem is perturbed from its present-day state by the physical ocean changes associated with unchecked anthropogenic emissions[14,25]. Over the course of the 21st century mean global sea surface temperature (SST) increases by 3 °C, there is increased stratification and reduced mixing at the surface. The meridional overturning circulation slows and shallows relative to current day conditions. These changes lead to a reduction in the supply of macronutrients from depth. Sea-ice retreats.

The shifts in Chl-a (Fig. 4a) reflect multiple physical and physiological changes. In many regions there is a decrease in Chl-a and productivity due to reduction in macro-nutrient supply. In polar regions a reduction in sea-ice leads to greater productivity as more sunlight reaches the surface ocean. In other mid to high latitude regions a complex combination of stratification-induced reduction of light limitation (impacting both growth rates and photo-acclimation[26]), decreased nutrient supply, and increased growth rates due to warmer temperatures, leads to a mixed pattern of positive and negative responses[13]. In general, Chl-a changes in the same direction as primary production, but subtle differences suggest that alterations in Chl:C ratios also play a role. In the model Chl:C ratios are driven by light, temperature, and nutrient stress[27]. As in most climate change models[11,13,14,28], larger phytoplankton that are disadvantaged in lower nutrient conditions decrease in biomass relatively more than smaller phytoplankton (Supplementary Fig. 5b). Smaller phytoplankton and diazotrophs increase their habitat range[13,18]. There is a significant shift in the total and relative abundances of the different phytoplankton types resulting in alterations to the community structure, measured here as a Bray–Curtis Dissimilarity index[29] (Fig. 4b, see Methods).

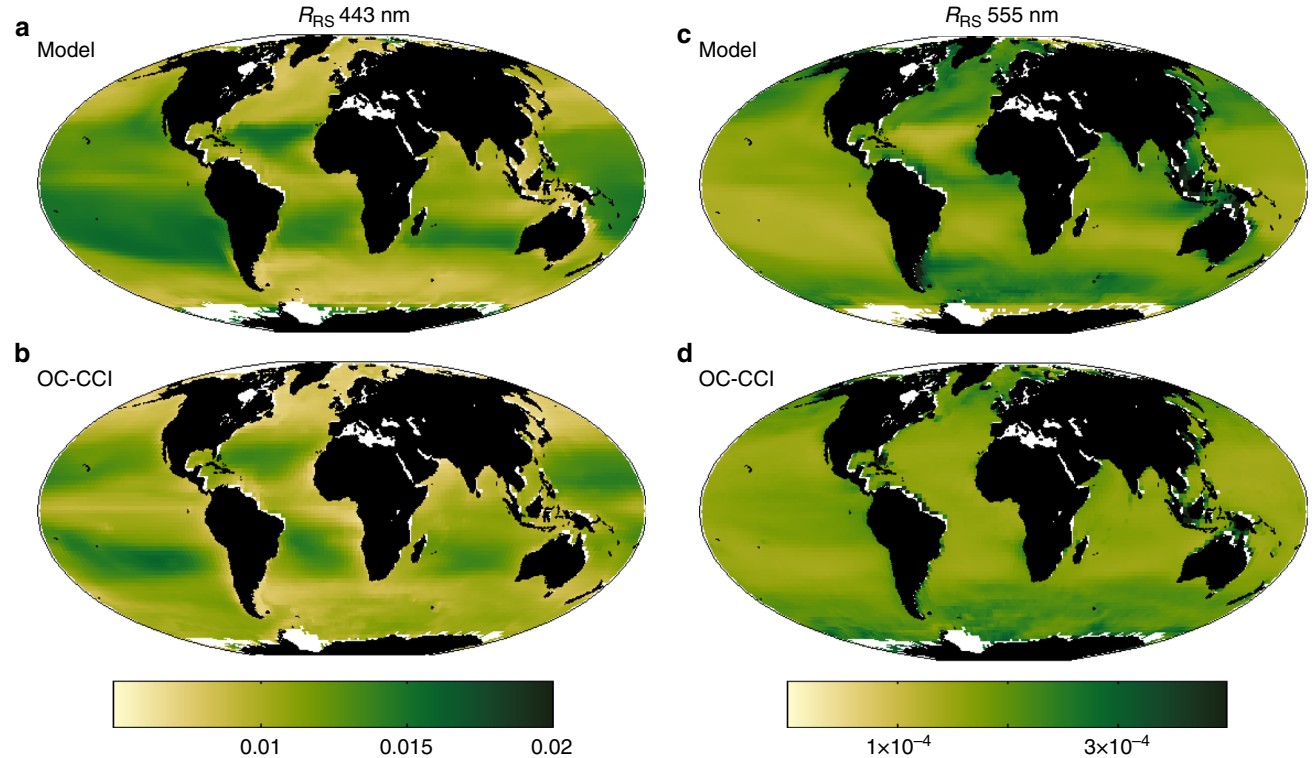

**Fig. 2** Remotely sensed reflectance. Current day composite (1998–2015) for **a** model $R_{RS}$ interpolated to 443 nm, **b** observed $R_{RS}$ at 443 nm, **c** model $R_{RS}$ interpolated to 555 nm, and **d** observed $R_{RS}$ at 555 nm. Units are $sr^{-1}$. Observed fields are from the Ocean Colour Climate Change Initiative (OC-CCI) project. White areas are regions where model resolution is too coarse to capture the smaller seas or regions of constant ice cover. Statistics of comparison of model and all six observed wavebands are provided in Supplementary Figs. 1, 2

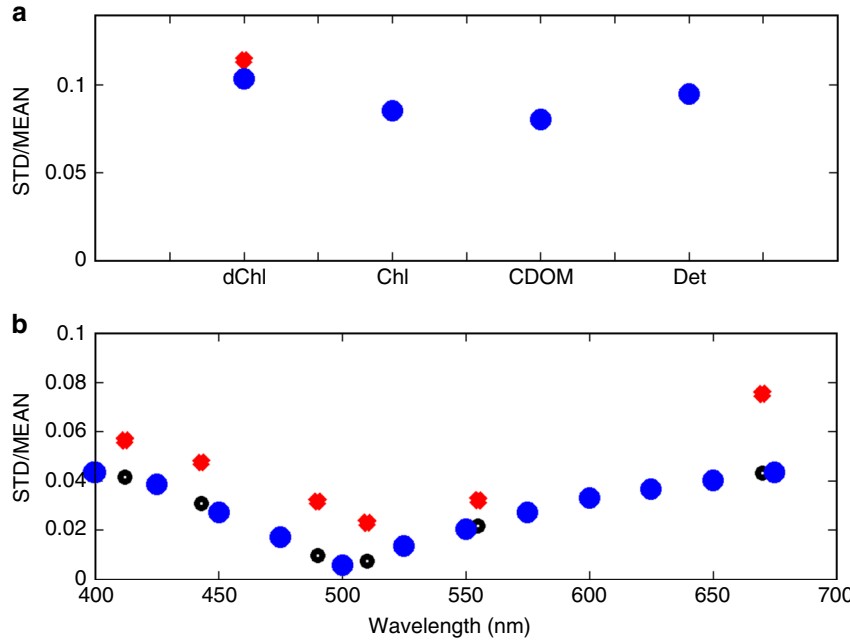

**Fig. 3** Relative magnitude of interannual variability. Global median of the ratio of the annual composite temporal standard deviation to the 18 year mean composite for **a** satellite derived Chl-a (dChl), actual Chl-a (Chl), detrital matter (det) and CDOM; and for **b** remotely sensed reflectance. Blue indicates model output, red for OC-CCI products, and black for the model interpolated to the OC-CCI wavebands. The OC-CCI reflectances are at 412, 443, 490, 510, 555, 670 nm. See Supplementary Fig. 4 for spatial patterns of relative magnitude or both model of OC-CCI products

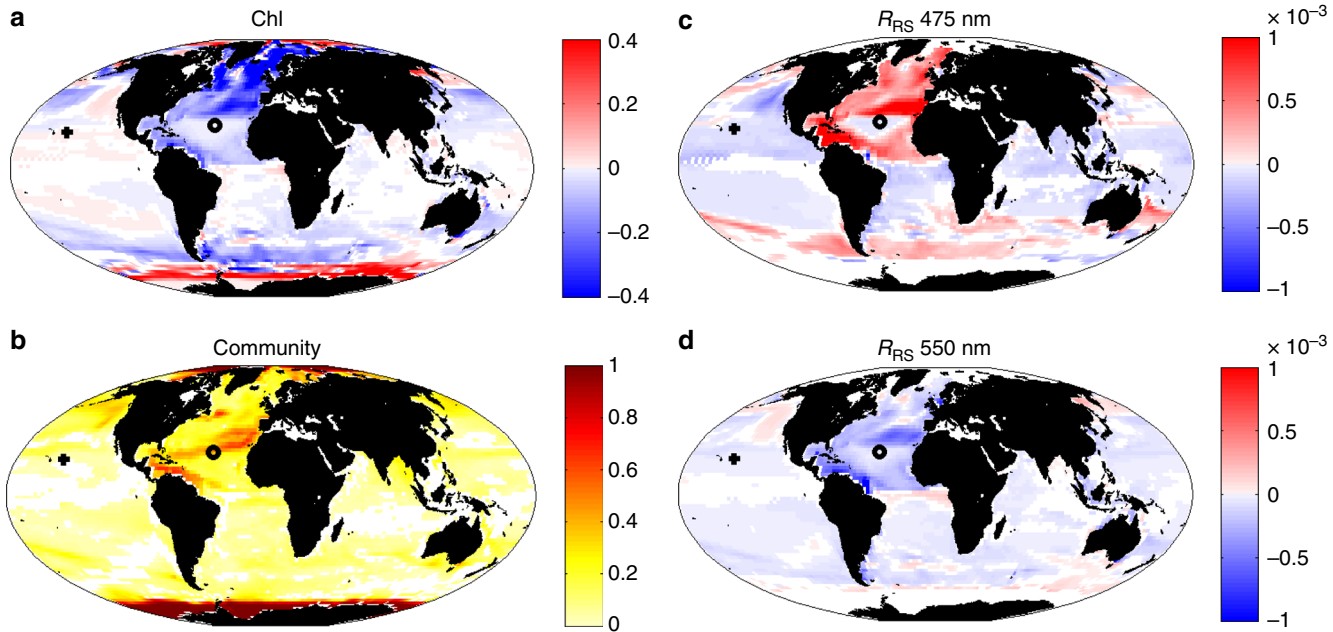

**Fig. 4** Change between current day and end of the 21st century. **a** Difference in model actual Chl-a (mg Chl m$^{-3}$) between 2085–2100 mean and the current day (1998–2015) mean. **b** Change to phytoplankton community structure as defined from Bray–Curtis dissimilarity index for community structure averaged over 2085–2100 versus the present day community (1998–2015). 0 indicates no change, 1 indicates a completely new community. Difference in model $R_{RS}$ 2085–2100 mean and the current day (1998–2015) mean for **c** 475 nm (blue) and **d** 550 nm (green). In all panels only areas with statistically significant differences between the two periods ($p < 0.05$) are shown. In addition, in **c** and **d** we only show regions which were ice free for most of the year (i.e. open ocean where $R_{RS}$ was calculated) in the current day. The symbols (+,o) indicate two locations highlighted in Fig. 8

Having established the main changes to the physics, biogeochemistry and ecosystem in response to the climate perturbation, and that they are consistent with previous studies, we now explore the associated changes in the ocean colour response.

**Optics and ocean colour response.** We find that the colour of the ocean will change. Here we use the hue angle ($\alpha$, see Methods) to quantify true colour (Fig. 5a). This metric uses the $R_{RS}$ spectrum together with the spectrum from the sensitivity of the human eye[30] to provide a value between 0° and 360°. Open ocean values range between green (~100°) where there is high productivity to blue (~230°) in the oligotrophic subtropical gyres (Fig. 5a). We find a change in the hue angle by up to 10° in some locations and a decrease of up to 5° in others (Fig. 5b). We note that these are relatively small shifts, unlikely to be easily registered by eye. Increase in the hue angle can be interpreted as a shift to bluer water, while a decrease suggests greener water. In general, the pattern matches that of the change in Chl-a (Fig. 4a): bluer water where there are decreases in Chl-a and greener waters where Chl-a increases. These results can be understood by the changes seen in the blue $R_{RS}$ (shown for 475 nm, Fig. 4c), which increases in most regions where Chl-a decreases. Chl-a absorbs strongly in the blue, such that a larger amount of blue light is reflected when there is lower Chl-a. The reverse occurs for most regions where Chl-a increases: increased blue reflectance and greener colour. There is less impact of the changes in Chl-a in the green waveband (Fig. 4d). This strong impact of Chl-a concentrations on the blue reflectance, but less on the green, is used by many algorithms that determine Chl-a concentrations from space[2] (see Methods).

Reflectance (and hence ocean colour) is determined by the total amount of absorption ($a_{tot}$) and backscattering (bb$_{tot}$) in the water: $R_{RS} \sim \frac{bb_{tot}}{a_{tot}+bb_{tot}}$. Absorption of irradiance in any waveband ($\lambda$) is the sum of the contribution of the main constituents (water

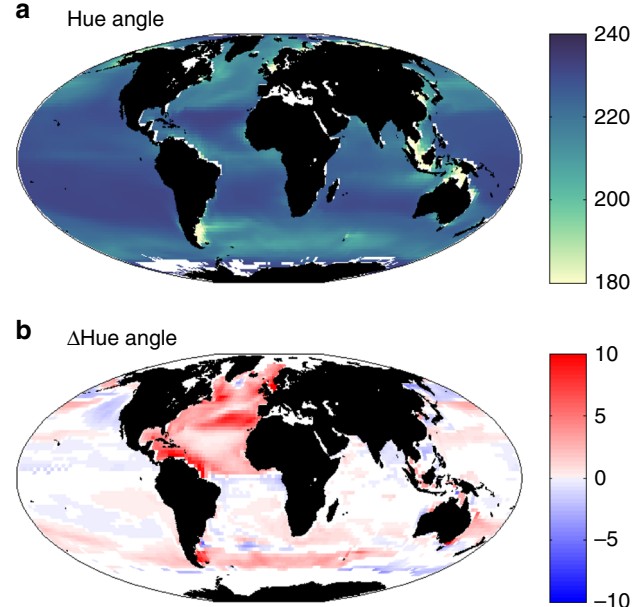

**Fig. 5** Hue angle. **a** Mean for 1998–2015, **b** difference in model 2085–2100 mean and the current day (1998–2015). In **b** only areas with a statistically significant differences between the two periods ($p < 0.05$) and which were ice free for most of the year (i.e. open ocean where $R_{RS}$ was calculated) in the current day are shown

molecules, phytoplankton, CDOM and detrital matter):

$$a_{tot}(\lambda) = a_w(\lambda) + a_{phy}(\lambda) + a_{cdom}(\lambda) + a_{det}(\lambda) \quad (1)$$

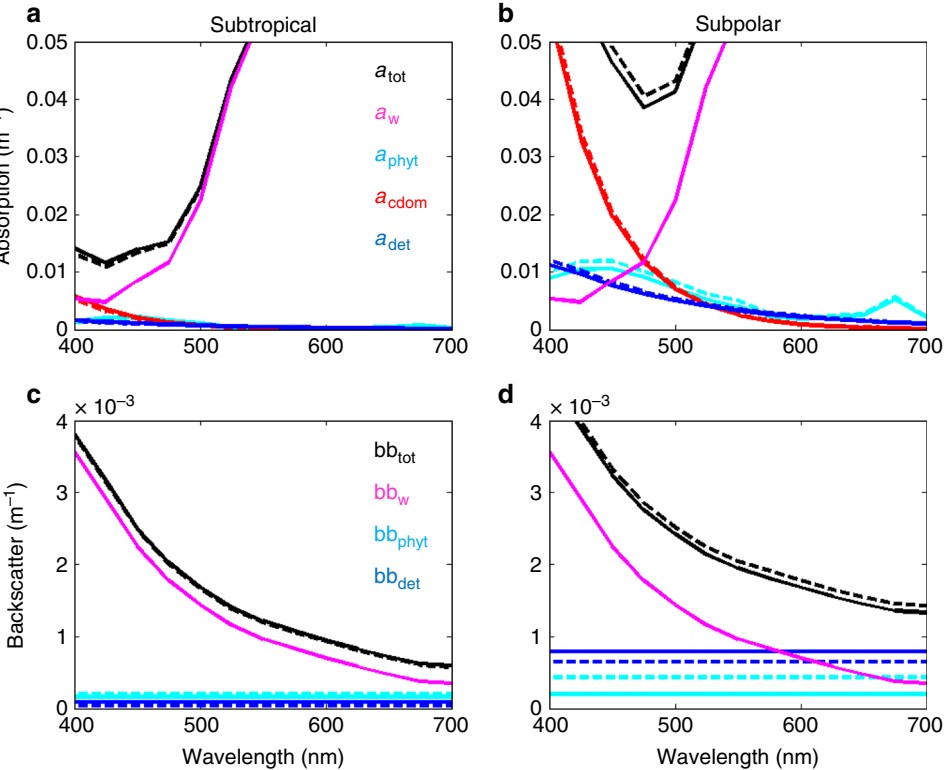

**Fig. 6** Absorption and backscatter. Components of total absorption (**a**, **b**) and backscattering (**c**, **d**) for two locations: middle of the oligotrophic subtropical gyre (circle in Fig. 4) (**a**, **c**) and the productive subpolar gyre in the North Atlantic (**b**, **d**). Units are m⁻¹. Solid lines are for current day (15 year mean) and dashed lines are for the mean of the last 15 years of the 21st century. Black is for total, purple for water, red for coloured dissolved organic matter (CDOM), light blue for phytoplankton, and dark blue for detrital particles

and backscatter:

$$\mathrm{bb_{tot}}(\lambda) = \mathrm{bb_w}(\lambda) + \mathrm{bb_{phy}}(\lambda) + \mathrm{bb_{det}}(\lambda) \qquad (2)$$

In general, the lowest absorption (and hence the most amount of light available for phytoplankton growth) occurs in the blue and blue-green portions of the spectrum between the strong absorption by CDOM and water[31] (Fig. 6a, b).

The amount of each constituent relative to each other impacts the ocean colour (i.e. the spectrum of light seen from satellite). The source of CDOM and detritus are modelled to increase/ decrease with higher/lower productivity. Over the course of the 21st century, the change in the relative roles of CDOM and detritus on absorption usually have the same sign as for Chl-a (Fig. 7). However, there are many regions where there are opposite responses or the relative change to each other are inconsistent in magnitude. This is because processes other than productivity impact the constituents differently. For example, CDOM is bleached by sunlight[32], and the combined lower production and increased bleaching due to higher stratification decreases the relative importance of CDOM to the absorption of irradiance in many regions (Fig. 7b). Larger phytoplankton cells are modelled to produce relatively more particulate matter than smaller cells, so as the community shifts to smaller cells we find a larger decrease in the importance of particulate detrital matter relative to Chl-a in many locations (Fig. 7c). Chl:C ratios are also altered by such changes in the community structure. Thus, in different regions of the ocean various combinations of relative changes occur depending on the local alterations to stratification, productivity, community structure and photo-acclimation, driving differing effects on reflectance. Here we have specifically examined $R_{\mathrm{RS}}$ at 475 nm, but other different effects also occur at

other wavebands and it is the combination of responses that leads to the overall changes in ocean colour (Fig. 5).

To understand how the trends in the biogeochemistry, ecology and optics, as simulated for the business as usual scenario, relate to current observing capabilities, we ask: When will these changes be unambiguous relative to natural interannual variability?

**Biogeochemical and ecosystem trends and time of emergence.** We first explore when the anthropogenic climate change signal exceeds the natural variability in Chl-a. We calculate the linear trend of Chl-a between 1995 and 2100 (Figs. 8a, c and 9a) using a generalized least squares fit (see Methods). In Fig. 9, we show only regions with statistically significant trends ($p < 0.05$). That the patterns in the trends (Fig. 9a) are similar to the differences shown in Fig. 4a, gives confidence that a linear trend analysis is appropriate (note that Fig. 4a shows absolute differences and Fig. 9a shows % trend, which magnifies the changes at lower Chl-a). The North Atlantic shows a strong negative trend (less than −0.5%/ year) and the North Pacific shows a weaker negative trend (alterations in limiting nutrient lead to a few regions showing a positive trend[13]). However, we find that many regions of the ocean do not have a statistically significant trend over the 21st century.

The time when the signal of climate change emerges from natural variability can be defined as the time of emergence[33–35]. Here we ask when the trend will be larger than the interannual variability of the present day (1998–2015). We define the natural variability as twice the standard deviation (STD) of the annual means at any grid location, such that the time of emergence (ToE) is:

$$\mathrm{ToE} = \frac{2 * \mathrm{STD}}{\mathrm{linear\ trend}} \qquad (3)$$

Only a few regions will show an unambiguous climate change signal in Chl-a before 2030 and many regions will not

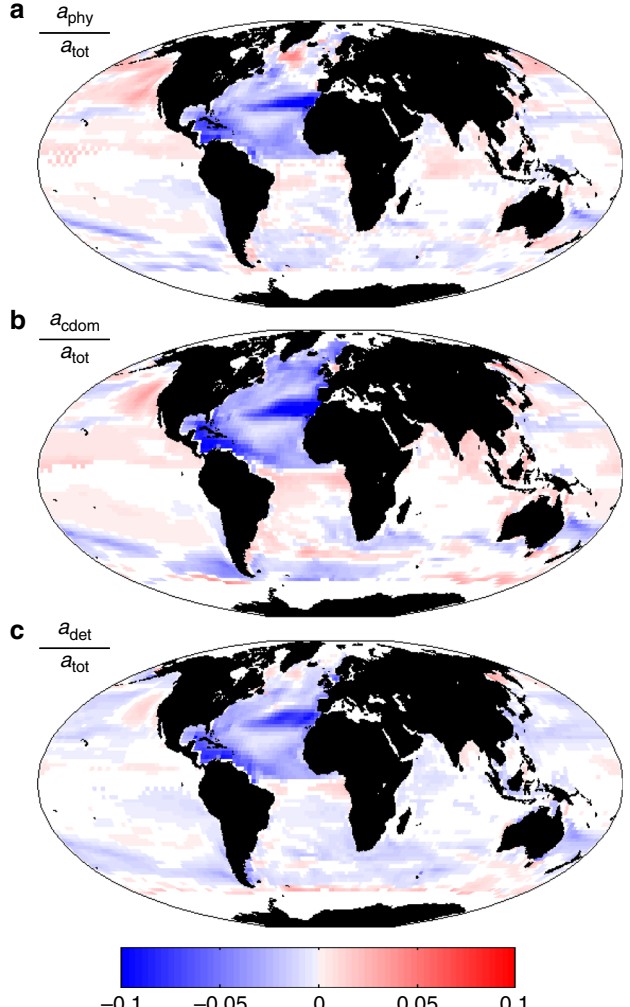

**Fig. 7** Change in contribution of optically important constituents. Change between 2085–2100 and 1998–2015 of **a** $a_{phy}/a_{tot}$; **b** $a_{cdom}/a_{tot}$; **c** $a_{det}/a_{tot}$ at 475 nm. Absorption is indicated by $a$: $a_{tot}$ refers to total absorption, $a_{phy}$ to the phytoplankton component of absorption, $a_{cdom}$ to the dissolved organic matter component, and $a_{det}$ to the detrital particle component. Only regions with statistically significant differences ($p < 0.05$) between the two sample periods are shown. In addition, we only show regions that were ice free for most of the year

Changes in phytoplankton community composition (as determined by Bray–Curtis Dissimilarity, Fig. 4b) have stronger and more significant trends over the course of the century (Fig. 10e) than any of the other metrics that we have considered to this point. A total of 50% of the ocean shows a signal of change by 2100, and even by 2040, 21% of the ocean has an unambiguous signal. In this model the community structure is determined by biogeochemical functional groups; in which differences between the groups are set by their nutrient requirements and roles in biogeochemical cycles (see Methods). These different functional groups also have distinct accessory pigments that lead to different absorption[36,37] and scattering spectra[36,38]. Thus, the unique combination of phytoplankton that coexist (the community) at any point in the ocean has a strong impact on the optics of the ocean[17,19,39,40] and changes to the communities will have an important optical signal.

Though these findings are informative about potential changes in the oceans, these are not as useful from the perspective of the properties that ocean colour satellites actually measure. Each of these quantities, i.e. actual Chl-a, CDOM, detritus and community composition, are quantities that can only be measured in situ. A more practical question would be: What is the percentage of the ocean area showing a significant trend in globally observable quantities at different intervals over the 21st century?

**Optics and ocean colour trends and time of emergence**. We consider how the model $R_{RS}$ manifests trends and time of emergence of the unambiguous climate change signal (Figs. 8b, d, 9b, d and 10 g). We find that the reflectance trend in the blue (475 nm) waveband (Fig. 9b) is mostly anti-correlated with the Chl-a trends (Fig. 9a) as already seen in the difference plots (Fig. 4). However, in some regions (e.g. central North Atlantic gyre) other optical constituents become more important and anti-correlation is not as clear (see e.g. Fig. 8a, b). In regions of the largest Chl-a trends, the 475 nm (blue) waveband $R_{RS}$ trend is >0.1%/year (or 1%/decade).

Globally, all wavebands of the model reflectance (Fig. 10g) have a stronger trend than the individual optically important water constituents (Fig. 10b–d) other than community structure changes. However, given that the model underestimates the natural variability especially in the high wavebands (Fig. 3b), we focus only on our results for the lower (<600 nm) wavelength bands.

The strongest signal is in the blue-green range (in our model the two 25 nm wavebands centred at 475 and 500 nm). These wavebands show a clear signal over 50% of the ocean by the end of the century. The strongest signal of trend was found over the wavelengths spanning from 487 to 512 nm, with 63% of the ocean providing a significant signal by 2100. These are the wavebands where there is the least interannual variability in the model (Fig. 3b, Supplementary Fig. 4). This matches what is found in the OC-CCI product where interannual variability is lowest in the 490 and 510 nm wavebands. These wavebands also lie on the edge of the wave-space between the strong absorption by CDOM in the bluest bands (Fig. 6) and where the impact of absorption by water starts to become more significant in the higher wavebands. These bands are likely to be most sensitive to changes in community composition. The low interannual variability together with sensitivity to changes in all water constituents suggest that these wavebands in the real world satellite sensors (490 and 510 nm) might be the first measurements to detect climate change signal in the marine ecosystem.

The metric of true colour, the hue angle, which is composed of the full visible reflectance spectra does not show a particularly strong signal of change (Fig. 10f). Where some of the reflectance

show a signal by 2100 (white areas in Fig. 9c). We can summarize these results by asking: What is the percentage of the ocean area showing a significant trend at different intervals over the 21st century? This analysis (Fig. 10b) suggests that <5% of the ocean has a significant trend in Chl-a by 2030, and only 31% by 2100. The high interannual variability in Chl-a (Fig. 3a) results in few regions having a sufficiently strong signal for the trend to be detectable (a similar conclusion was found in a previous study[15]). However, does this mean that there are only small changes occurring in the ocean ecosystem and biogeochemistry over the course of the 21st century in this scenario? We consider other optically important constituents of the water.

Our model suggests that detrital matter will show an even slower and less obvious signal than Chl-a (Fig. 10c), but that CDOM will have a stronger signal (Fig. 10d) with 36% of the ocean showing a significant trend by 2100. The increased bleaching of CDOM in the more stratified surface waters leads to this stronger signal relative to Chl-a.

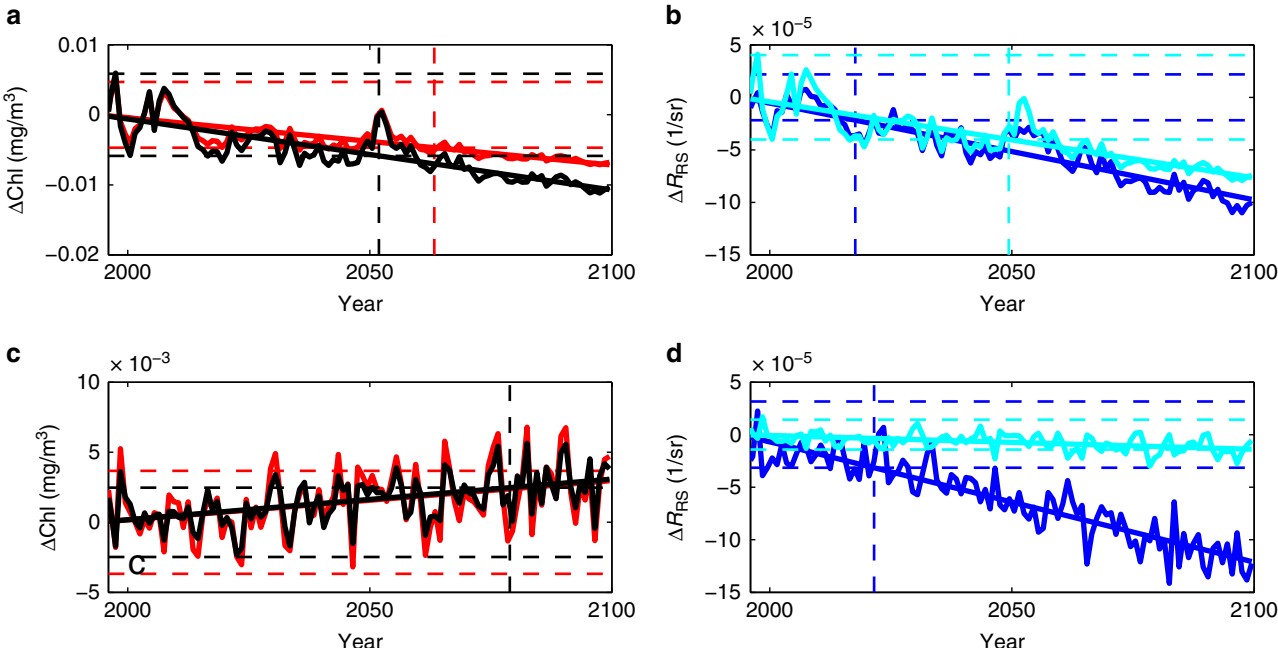

**Fig. 8** Time series of changes in two locations. **a**, **b** are in the North Atlantic, indicated by circle in Fig. 4. **c**, **d** are in North Pacific, indicated by cross in Fig. 4. **a** and **c** show the changes in model actual Chl-a (black) and model Chl-a product derived from reflectance ratio (red); **b**, **d** show changes in $R_{RS}$ for 475 nm (dark blue) and 550 nm (light blue). Straight solid lines indicate the linear trend using generalized least squares (GLS), the dashed horizontal lines indicate plus and minus two standard deviations (STD) of the interannual variability from 1998 to 2015, and the vertical dashed line shows the time of emergence (trend > twice the STD)

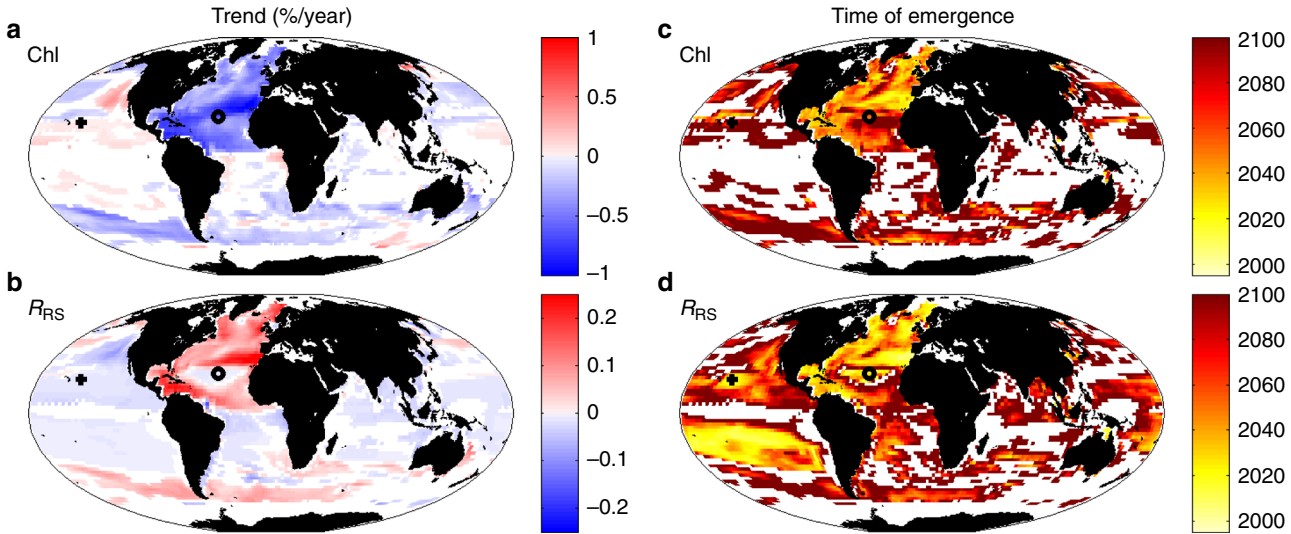

**Fig. 9** Trends and time of emergence. Model linear trend (%/year) for **a** actual Chl-a, and **b** remotely sensed reflectance at 475 nm; and time of emergence of trend for **c** Chl-a, and **d** $R_{RS}$ at 475 m. A generalized least squares (GLS) fit was used to quantify the trends. Only regions with statistically significant ($p <$ 0.05) trends over the 21st century and that were largely ice-free in the current day (as model $R_{RS}$ are only valid for such regions) are shown. The symbols (+,o) indicate two locations highlighted in Fig. 8

bands do not show a strong trend, they will dampen the signal of the change in true colour. While surface ocean colour harbours important information on the ocean ecological and biogeochemical response to climate change our results indicate that it is necessary to resolve blue-green bands (nominally 467–510 nm) to obtain the most sensitive indicator. This is not due to overall changes in Chl-a per se but rather due to combined changes in optical constituents, and particularly phytoplankton community structure.

**Consequences for satellite-derived Chl-a**. We further ask how the shifts in relative importance of different constituents will affect ocean colour products (specifically Chl-a) that are derived from these reflectance measurements. A particular concern is that the algorithm coefficients used for the contemporary ocean may not generate accurate Chl-a estimates for an optically different future ocean.

We explore how the model-derived Chl-a and actual model Chl-a differ over the course of the 21st century (Fig. 8a, c). In general,

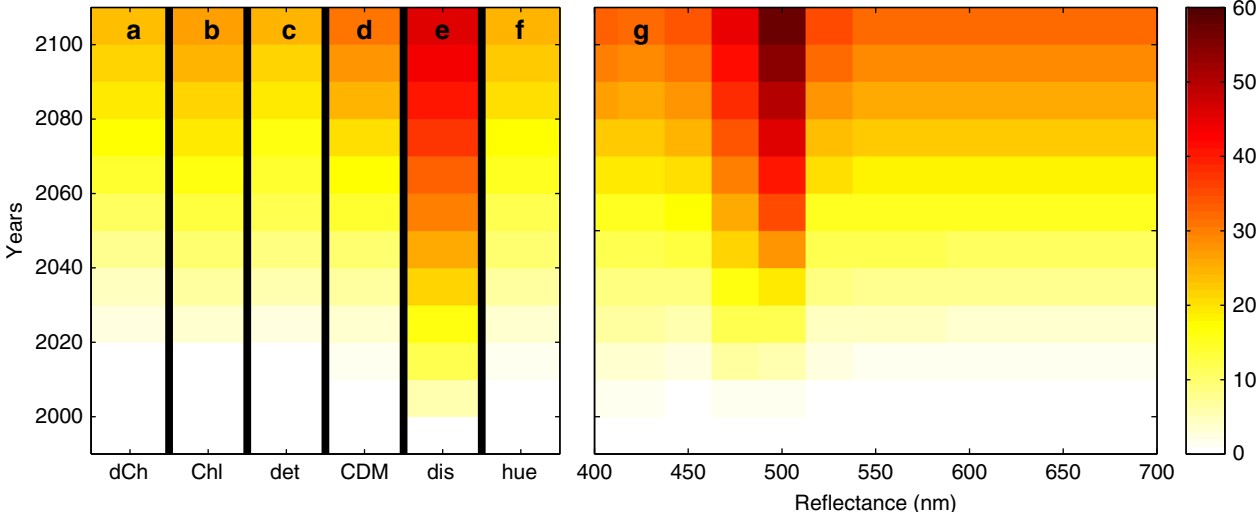

**Fig. 10** Percent ocean area with significant trend. The amount of open ocean area that has statistically significant trend ($p < 0.05$) since 1995 with 5-year increments. **a** Satellite-like derived Chl-a is the blue/green reflectance ratio Chl-a product calculated with an OC4-like algorithm, **b** actual Chl-a is the sum of the 8 dynamically changing phytoplankton which contribute to total Chl-a (as might be measured in situ); **c** detritus is the non-living particulate organic pool, **d** CDOM is the coloured dissolved organic matter, **e** dis refers to Bray–Curtis Dissimilarity index, and is a measure of the changes to the phytoplankton community structure; **f** hue refers to the hue angle, $\alpha$, a metric of true colour; **g** the remotely sensed reflectance in the visible spectrum, in 13 wavebands

the trends in the two have the same sign, but in over 13% of the ocean the trend in derived Chl-a is under-estimated relative to the trend in actual Chl-a (e.g. Fig. 8a) and in 15% it is over-estimated. Note that only 28% of the ocean has a statistically significant trend in the model-derived Chl-a. Thus, the changes in the relative importance of the different constituents suggests that the algorithms developed for satellite derived Chl-a for the present day will not necessarily be valid over the course of the 21st century. This reduced ability to capture trends successfully, as well as differences in the variability in the derived product in some locations (see e.g. Fig. 8c), also leads to the derived Chl-a having an even smaller signal of change and fewer regions showing statistically significant long-term trends than the model actual Chl-a (Fig. 10a).

We note that there is a continuation of algorithm development[41,42], with current products often using a blend of algorithm types depending on the region or water optical type (see e.g. the NASA Reprocessing 2014.0, and OC-CCI V3 release). However, algorithms based on reflectance ratios still use current day datasets to fit coefficients. We suggest that algorithm coefficients will need to be continuously adjusted based on subsets of newer in situ measurements as the optical properties of the ocean alter. On the other hand, there are also developments in semi-empirical inversion algorithms[3,4] that also estimate the contributions of CDOM, particle backscattering, as well as Chl-a. As such, these techniques may be less affected by the changing relative importance of the different water constituents.

**Uniqueness and caveats of the model.** The results presented in this manuscript must be interpreted carefully, i.e. in the context of the ocean ecosystem and optical models simplified representation of the real world. Though relatively complex, the model still has only a limited number of optically different plankton species and does not include several important optical constituents (e.g. viruses, minerals[36]) or the effects of salinity[20] that could impact the accuracy of the reflectance and also the ability to capture the natural interannual variability. The parameterization of CDOM and detrital matter is still simplistic, and we caution that the model cannot yet be used to make definitive comments on these changes, but can provide a unique chance to explore the potential

alterations. The patterns and magnitudes of Chl-a and $R_{RS}$ match satellite products (Figs. 1 and 2, Supplementary Figs. 1, 2), and the relative magnitude of interannual variability is also captured (Fig. 3, Supplementary Fig. 4). However, there are discrepancies, with the model overestimating Chl-a in high latitudes and suggesting a much more limited region of variability in the Equatorial Pacific than observed. The interannual variability of $R_{RS}$ at higher wavebands is also underestimated. These discrepancies suggests that the model is still missing important processes and constituents that affect variability. The model has relatively coarse (25 nm) wavebands, which differ in size and spacing from historic and current satellite sensors. Thus, the model can only provide broad estimates of the wavelengths, and cannot in its current form suggest bandwidths, that will encompass the strongest long term change signals. However, despite these caveats, the model provides a unique opportunity to investigate trends in ocean colour over the 21st century, and identify which signals will have the strongest response to anthropogenic climate change. Very few models currently include a radiative transfer component or simulate products such as reflectance[17,43–45], and no model with these capabilities has looked at a climate change scenario.

## Discussion

We use a model that includes ocean physics, biogeochemistry and ecosystem components, plus an explicit representation of the water optical properties and radiative transfer to consider the ocean colour signature of climate change. We consider a business-as-usual (similar to IPCC RCP8.5) scenario of greenhouse gas emissions over the course of the 21st Century, which leads to substantial changes to the physics, biogeochemistry and ecology of the ocean[13,14,18]. We ask: How do the colour and optical nature of the ocean respond to climate change, and how long will it take for an unambiguous signal of climate change to emerge in ocean colour properties?

We find that the most widely used indicator of marine phytoplankton, Chl-a, does not evince rapidly detectable long-term trends, due to large natural variability. Instead, our study suggests that individual wavebands of colour (here the $R_{RS}$) have a more

readily detectable climate change trend, over a greater proportion of the ocean, than the hue angle, Chl-a, CDOM or detrital matter. For instance, $R_{RS}$ centred at 500 nm has a significant trend by 2100 over 63% of the ocean, compared to only 31% of the ocean for Chl-a. Reflectance has lower natural variability than Chl-a and other in-water properties (Fig. 3). However, $R_{RS}$ also integrates the alterations of all the optically important constituents. These two aspects lead reflectance to be a more sensitive indicator of change than any single constituent by itself. Changes to reflectance might provide the earliest alert from ocean colour satellites of climate change impacts on the bulk marine ecosystem. However, the precision and drift in ocean colour sensors will need to be sufficiently low to capture these signals (trends generally <1% per decade).

Though trends in $R_{RS}$ will not necessarily identify specific changes (e.g. change in CDOM versus community structure), they will indicate that there are ecosystem-level alterations taking place. Community structure changes impact the water optics since phytoplankton types absorb and scatter light differently to one another[36]. Though more difficult to monitor from space (though see IOCCG report 15[46]), our study suggests that changes in phytoplankton community structure will have a stronger climate-change-driven signal over the course of the coming century than Chl-a. Sustaining time series measurements that include information on taxonomic or functional group biomass is therefore highly desirable, as is the continued improvement in our ability to detect phytoplankton diversity from space[39,40,46,47]. Changes in the types of phytoplankton in any location could have dramatic impacts on higher trophic levels if grazers and their predators cannot switch to the new community.

We find that the optically important water constituents do not all alter in the same manner. For instance, CDOM is increasingly bleached with higher stratification and community shifts alter the relative amount of detrital matter. The shifts in the relative importance of the optical constituents alter the spectral makeup of the water leaving irradiance and the ocean colour. Such a change will pose a problem to any of the ocean colour products that utilize in situ knowledge of the ocean's current optical make up. We show that Chl-a derived from $R_{RS}$ with a reflectance ratio algorithm developed for the current day does not capture the correct trends over the course of the 21st century. New algorithm development need to keep the ocean's altering optical properties in mind. Shifts in optical properties should be taken into account when studying trends in Chl-a and productivity in the real ocean, even in the near term. Such studies might consider time-varying algorithms. Techniques that separately estimate the different water constituents[4,42] may provide a better avenue for detecting trends.

There is considerable effort expended to determine the best spectral bands for satellite sensors[40,48], and numerical models are starting to be used to explore aspects of band selection for future missions (e.g. PACE)[43]. We suggest that the choice of bandwidths for future satellite sensors should also include estimates of the strength of trends they will capture. Our work identifies 467–512 nm as promising from this aspect. Current and historic sensors (e.g. SeaWiFS, MODIS, VIIRS, MERIS) have all included wavebands around 490 nm and we suggest that it is imperative to maintain a similar band in future missions for the earliest signatures of marine ecosystem changes. Our results also suggest that including sufficient bands to detect different communities (e.g. hyperspectral) and those that will help separate signals of CDOM (e.g. ultra-violet) will be important for monitoring ecosystem changes (such bands are planned for the PACE mission).

The estimates of bulk ocean phytoplankton, such as Chl-a, commonly used in assessing the influence of both natural variability and climate change on marine ecosystems mask more nuanced changes to the community structure and in turn their impact on other optically important constituents of the ocean. Although overall ecosystem productivity may change more slowly, relatively rapid changes to phytoplankton community structure may have significant knock-on effects for the fecundity and composition of the local higher trophic level community.

Our results thus suggest several focus areas important for monitoring the response of ocean productivity to climate change: maintaining ocean colour sensor compatibility and long term stability, particularly in the 490 nm waveband; maintaining long-term in situ time-series of plankton community, e.g. the Continuous Plankton Recorder survey and repeat stations (e.g. HOT, BATS); reducing uncertainties in satellite-derived phytoplankton community structure estimates.

## Methods

**The numerical model**. We use the biogeochemical/ecosystem/optical numerical model of ref. [17] coupled to the MIT Integrated Global System Model (IGSM[13,14,18,25,49–53]), an integrated assessment model that links an Earth system model of intermediate complexity to a human activity model. We provide a brief description here, including the pertinent features, but refer the reader to the above cited papers for more details, equations, parameter values and evaluation.

The marine biogeochemical component resolves the cycling of carbon, phosphorus, nitrogen, silica, iron, and oxygen through inorganic, living, dissolved and particulate organic phases. The ecosystem component resolves 8 phytoplankton types (diatoms, coccolithophores, pico-eukaryotes, *Synechococcus*, high and low light *Prochlorococcus*, *Trichodesmium* and unicellular diazotrophs) and two grazers. The phytoplankton types differ in the nutrients they require (e.g. diatoms require silica), maximum growth rate, nutrient half saturation constants, sinking rates, and palatability to grazers. The phytoplankton also differ in their spectral absorption and scattering characteristics (see Fig. 1 in ref. [17]) and maximum Chl-a:C ratio. Chl-a:C varies as a function of the light, temperature and nutrient environment[27].

The model includes explicit radiative transfer of spectral irradiance in 25 nm bands between 400 and 700 nm. The three stream (downward direct, $E_d$, downward diffuse, $E_s$, and upwelling, $E_u$) model follows previous studies[54–56], though here it is reduced to a tri-diagonal system that is solved explicitly[17]. The model simulates the spectral absorption and scattering properties of water molecules, the 8 phytoplankton types, detritus and coloured dissolved organic matter (CDOM). Irradiance just below the surface of the ocean (direct, $E_{d0}$, and diffuse, $E_{s0}$, downward) is provided by the Ocean-Atmosphere Spectral Irradiance Model (OASIM[56,57]). See ref. [17] for more details.

The marine biogeochemical and biological tracers are transported and mixed by the MIT general circulation model (MITgcm[58]), the three-dimensional ocean component of the IGSM. The ocean component has a 2° × 2.5° resolution in the horizontal, and twenty-two layers in the vertical, ranging from 10 m at the surface to 500 m thick at depth[13,14,18,52]. The Earth system model in the IGSM also includes a simplified representation of atmospheric dynamics, physics and chemistry, along with terrestrial water, energy and ecosystem processes, and a full carbon cycle[49,52,53]. The ocean physics displays a realistic year-to-year variability in surface temperature and produces interannual variability (e.g. ENSO) with frequency, seasonality, magnitude and patterns in general agreement with observations[52,53].

In this study, because of the high computational demand of the biogeochemical/ecosystem/optical numerical model, we use a single climate simulation from an ensemble of perturbed physics (climate sensitivity), perturbed initial conditions, and varied emissions scenarios. We focus on the climate simulation with a medium climate sensitivity (3.0 °C) under a business-as-usual scenario similar to the Representative Concentration Pathway 8.5 (RCP8.5) used in the Coupled Model Intercomparison Project 5 (CMIP5)[24]. The coupled system is spun up for 2000 years (using 1860 conditions) before simulating 1860 to 2100 changes. Observed concentrations of greenhouse gases, ozone and aerosols, including volcanic stratospheric aerosols, as well as solar irradiance are used to force the IGSM from 1860 to 2000, and 21st century climate simulations are driven by anthropogenic emissions simulated by the human activity model.

Nutrient distributions were initialized from results from previous simulations, though the results presented here were not sensitive to these initial conditions. The ecosystem was forced with the physical fields from the Earth System Model for the pre-industrial control and run for 50 years to allow the phytoplankton community and the upper ocean biogeochemistry to establish a quasi-equilibrium. A repeating seasonal cycle was quickly reached and there was only a small biogeochemical drift associated with upwelling of deep water. The several thousand years of integration needed to adjust the deep ocean was computationally unfeasible with the full ecosystem model. After the 50-year spin-up, the transient run from 1860 to 2100 was performed. A second simulation was conducted with no increase in greenhouse gas emissions. This control simulation showed that there were no significant drifts in the ecological or optical properties discussed in this study.

The surface spectral irradiance was provided by OASIM[56,57] products, and the monthly surface iron dust fluxes were supplied by an atmospheric transport model[59]. These latter two fields were climatological means and did not change in the simulations described here. Though the impact of changes in light and dust are likely to be important in the future, they are beyond the scope of this paper.

**Model output of remotely sensed reflectance.** Importantly for this paper, the numerical model provides spectral surface upwelling irradiance: output that is similar to measurements made by ocean colour satellites. Only a few biogeochemical/ecosystem models have the ability to capture this diagnostic[17,43,45], and until now such diagnostics have not been part of a climate change simulation. We follow the procedure as discussed in ref. [19] to calculate $R_{RS}$. We calculate model reflectance for each waveband as the upwelling just below the surface ($E_u$) divided by the total downward (direct and diffuse) irradiance also just below the surface (as provided by OASIM): $R(\lambda, 0^-) = \frac{E_u(\lambda)}{E_{d0}(\lambda) + E_{s0}(\lambda)}$. We first convert model subsurface irradiance reflectance to remotely sensed reflectance just below the surface using a bidirectional function $Q$: $R_{RS}(\lambda, 0^-) = \frac{R(\lambda, 0^-)}{Q}$. The bidirectional function $Q$ has values between 3 and 5 sr[60] and depends on several variables, including inherent optical properties of the water, wavelength, and solar zenith angles[60,61]. Here for simplicity we assume that $Q = 3$ sr (as done in refs. [19,43]). Secondly, we convert to above surface remotely sensed reflectance using the formula of Lee et al.[62]: $R_{RS}(\lambda, 0^+) = \frac{0.53 R_{RS}(\lambda, 0^-)}{(1 - 1.7 R_{RS}(\lambda, 0^-))}$. Hereafter we will refer to this quantity as $R_{RS}$ which has units of sr$^{-1}$ and is equivalent to the $R_{RS}$ provided by ocean colour satellite databases.

**Deriving Chl-a product from remotely sensed reflectance.** The ocean colour Chl-a product that is most frequently used is based on the blue/green reflectance ratio (e.g. NASA OC4 algorithm for SeaWiFS and OC-CCI, version2). This product uses the fact that phytoplankton absorb more in the blue range of the light spectrum than the green. The ratio of the amount of blue to green light reflected at the ocean surface at any location therefore supplies information on the concentration of Chl-a. In particular, a 4th order polynomial can be constructed to estimate Chl-a from measured blue/green reflectance ratios[2]:

$$chl_d = 10^{a_0 + a_1 X + a_2 X^2 + a_3 X^3 + a_4 X^4} \quad (4)$$

Here $X = \log(R_{RSB}/R_{RSG})$, where $R_{RSB}$ is blue reflectance and $R_{RSG}$ is the reflectance in the green range. Typically, in the real world, the values of coefficients $a_0$ to $a_4$ are found using datasets of coincident in situ radiometric and Chl-a measurements. This empirical algorithm is then applied globally with satellite remotely sensed reflectance. Our recent study[19] shows that this method can also be used with model $R_{RS}$ output, creating a credible ocean colour like Chl-a product. We follow their approach, finding the coefficients using the model output for the current day, subsampled in space and time as is currently available for the real ocean in situ measurements[63]. $R_{RSB}$ is the blue reflectance (450 nm, 475 nm, or 500 nm, whichever is largest) and $R_{RSG}$ is the green reflectance (550 nm).

We note that there is considerable effort to improve the derived Chl-a algorithms in the ocean colour community[41,42]. Newer Chl-a products from NASA and OC-CCI use different algorithms in different regions of the ocean (e.g. low/high Chl-a or optically different provinces). Here, for simplicity we have focused on the simpler blue/green ratio OC4 algorithm. However, there are other approaches (e.g. semi-analytical inversion)[3,4] that attempt to more mechanistically estimate not only Chl-a concentration, but other constituents such as CDOM. We believe that exploring whether this approach will allow climate change trends to be more rapidly or robustly detected will be a promising avenue for future study.

**Model evaluation.** The ecosystem model has been evaluated in several recent papers[13,14,17–19]. Here we additionally show how the model derived Chl-a and remotely sensed reflectance ($R_{RS}$) compares favourably to the Ocean Colour Climate Change Initiative project (OC-CCI, https://www.oceancolour.org/) products (Figs. 1–3, Supplementary Figs. 1–4). We compare mean composites (and the interannual variability of these composites) over 1998 to 2015. Composites are derived from all monthly means where there are satellite measurements. Thus for instance, high latitudes only have input for months where there is sufficient light and some equatorial regions miss months when there are too many clouds. Model composites are derived with the same missing months to match the observations. The model internal interannual variability does not match the real world (i.e. El Ninos do not occur in the same years), thus we compare interannual variability in terms of a temporal standard deviation of annual composites from 1998 to 2015 (Fig. 1, Supplementary Fig. 3). To evaluate the skill of the model, we compare to satellite observations, which constitute only an estimate of true Chl-a, and $R_{RS}$ with potential uncertainties due to atmospheric corrections[1]. It must also be noted that the potential presence of discontinuities due to merging measurements from different sensors in the satellite record may also bias comparison with the model[64]. Indeed, the model does not contain such discontinuities and measures of agreement between the model and observations, such as correlation and relative bias (Supplementary Figs. 1–4), may underestimate how the model captures central tendency and interannual variability in the observations.

We show model actual and model satellite-like-derived Chl-a relative to the OC-CCI-derived Chl-a product (Fig. 1). The model-derived Chl-a (Fig. 1b) closely captures the actual Chl-a (Fig. 1a), though slightly overestimates the equatorial Chl-a. Comparing the model derived Chl-a to the OC-CCI product (Fig. 1, Supplementary Fig. 2), we find that the model captures the patterns of high and low Chl-a values between upwelling high latitude, equatorial and nutrient limited subtropical zones. The model has Chl-a too high relative to the OC-CCI product in high latitudes (sometimes by a factor of 2 or more), and there are some patches of the subtropical gyres that are too low. Regions of the equatorial Atlantic and Indian oceans are too high in the model. The region of high Chl-a in the equatorial Pacific is narrower in the model than the observations. Some additional regions of high productivity are not captured in the model, especially along coastlines where the model resolution is too coarse to capture the important coastal physics. The model also does not capture the polar regions well, either in the physics or sea-ice extent and we do not adequately parameterize sea-ice and sea-ice edge phytoplankton communities. The model also captures the patterns of interannual variability (Fig. 1e, f, Supplementary Fig. 3), but does overestimate it in regions where it also overestimates the 18 year mean composite. Particularly noticeable is that the high variability in the equatorial Pacific is shifted relative to the observations, suggesting the physical manifestation of the El Nino/La Nina response is slightly misplaced in the Earth System Model (unsurprisingly given the too narrow upwelling band as seen in the Chl-a composite). The model also does a good job at capturing the patterns of the ratio of the interannual variability to the 18-year composite (Supplementary Fig. 4), though overestimates in the Equatorial Pacific (where there is a mismatch in the physical manifestation of El Nino/La Nina), but otherwise has a low bias elsewhere. This latter is expected as we do not capture all the sources of variability (e.g. mesoscale features) found in the real ocean.

We further evaluate the model $R_{RS}$ (Fig. 2, Supplementary Figs. 1–4) in a similar manner to Chl-a. The model $R_{RS}$ are linearly interpolated from the 25 nm bands to the same bands as the OC-CCI product. The model captures the reversed patterns between blue (412,443 nm) and green (555 nm) $R_{RS}$ between gyres and highly productive regions. Several of the model biases reflect the biases seen in the Chl-a: underestimation of the blue band in the subtropics where modelled Chl-a is too low relative to the real ocean and higher green $R_{RS}$ in the equatorial Atlantic and Indian Ocean than the satellite measurements where the model overestimates Chl-a relative to the satellite product (Fig. 1, Supplementary Fig. 2). Additionally, the wavebands all show discrepancies in the Equatorial Pacific where the modelled Chl-a is too narrowly confined to the equator. The effect of salinity on $R_{RS}$ is not captured in the model, likely leading to additional discrepancies especially in the low latitudes where effects of salinity become more important[20]. Some equatorial regions (especially the Atlantic) also have high cloud cover and, in such regions, the satellite product may be biased. Correlations between model and OC-CCI (Supplementary Fig. 1) are best for the low and high wavebands, and worst for the blue-green (490 and 510). These latter are the wavebands where the reversal of the patterns of high/low $R_{RS}$ occur (see Supplementary Fig. 2) and are thus most difficult to capture correctly. For the same reason, these are the wavebands where the linear interpolation from the model bands to those of the OC-CCI products are most problematic. In general, the model has a low bias (Supplementary Fig. 3) for the interannual variability in $R_{RS}$, with the most noticeable exception in the equatorial Pacific where the Chl-a mismatch bias occurs. The model also captures the patterns of the ratio of the interannual variability to the 18-year mean composite (Supplementary Fig. 4). The major exception, again, is the equatorial Pacific where we have already noted the mismatch in placement of the interannual variability in Chl-a. The model is biased low in most other regions. However, the model does capture the lower ratio of interannual variability relative to the mean in the 490 and 510 nm than the other bands (Fig. 3, Supplementary Fig. 4).

**True colour classification.** We use the $R_{RS}$ to classify the true colour (the colour of light from the ocean which our eyes might capture) of the model ocean. Colour can be represented in terms of the three primary colours: blue, green and red. The sensitivity of the human eye to these primaries can be given by the colour matching functions[30]. We use these functions to convert the light spectrum from the ocean into three chromatic coordinates. The hue angle is a metric that compresses the spectrum in these three chromatic coordinates into a single value. Following van der Woerd and Wernard[65] we represent the three tristimulus values ($X$,$Y$,$Z$) in terms of the $R_{RS}$ spectrum:

$$X = I \sum_{i=400}^{700} R_{RS}(\lambda) \bar{x}(\lambda) \Delta\lambda \quad (5)$$

$$Y = I \sum_{i=400}^{700} R_{RS}(\lambda) \bar{y}(\lambda) \Delta\lambda \quad (6)$$

$$Z = I \sum_{i=400}^{700} R_{RS}(\lambda) \bar{z}(\lambda) \Delta\lambda \quad (7)$$

where illumination $I$ is taken as unity and $\bar{x}(\lambda)$, $\bar{y}(\lambda)$, $\bar{z}(\lambda)$ are the colour matching functions, and $\lambda$ is the wavelength at the middle of the waveband $\Delta\lambda$. We then express two coordinates ($x$,$y$) as: $x = \frac{X}{X+Y+Z}$, $y = \frac{Y}{X+Y+Z}$.

The hue angle is defined relative to white ($x_w = y_w = 1/3$) as:

$$\alpha = \arctan(y - y_w, x - x_w) \text{ modulus } 2\pi \qquad (8)$$

For the figures and discussion here we convert from radians to degrees. Typical ocean values of $\alpha$ go from about 40° for turbid brown waters in estuaries to over 220° in oligotrophic gyres[65].

**Changes in phytoplankton community structure**. As a metric of changes in community structure we use the Bray–Curtis Dissimilarity index[29]. Here we define the index $C_{t_i t_0}$ at time $i$ ($t_i$) as:

$$C_{t_i t_0} = 1 - \frac{2 \sum_{j=1}^{j=n} \min(B_{jt_0}, B_{jt_i})}{\sum_{j=1}^{j=n} B_{jt_0} + \sum_{j=1}^{j=n} B_{jt_i}} \qquad (9)$$

Where $B_j$ is the biomass of phytoplankton functional type $j$, $t_0$ is the mean from 1998 to 2018. The time period for $t_i$ is chosen based on the question to be addressed. For Fig. 4b, $t_i$ is a 15-year mean 2085–2100. To calculate trends, ToE, and provide output for Fig. 10e, we used annual average biomass for each year to construct yearly indices from 1995 to 2100. If there is no change in the community structure, the index will be 0. If there is a completely new community structure (i.e. no biomass of any of the original types), the index is 1. This index is calculated for each grid cell in the model.

**Trend analysis**. Given the strong autocorrelation of the residuals from an ordinary least squares fit, we instead used a generalized least squares fit[66,67] to find the trends in the different components. We used the R[68] function gls using annual means of the different fields from 1995 to 2100.

**Code availability**. The MITgcm model code is available through [https://mitgcm.org], code modifications specific to this simulations are available on Harvard dataverse: [https://doi.org/10.7910/DVN/UE8OS1].

## Data availability
The model output used for this study are available through Harvard dataverse: [https://doi.org/10.7910/DVN/08OJUV].

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

## Acknowledgements
We are grateful to Marcel Wernand for the suggestion to consider the hue angle changes in our model. S.D. and O.J. received funding from NASA (grant NNX16AR47G) and S.D. and E.M. from DOE (grant DE-FG02–94ER61937). C.B. was supported by a Marie Curie FP7 Reintegration Grants within the Seventh European Community Framework (project 631466—TROPHYZ).

## Author contributions
S.D., A.H., S.H., C.B. co-wrote the paper, with input from all authors. S.D. conceived the experimental design, conducted the biogeochemical/ecosystem/optical model simulations, and performed most of the analysis. O.J. was responsible for the numerical code and provided context for the optical diagnostics. E.M. conducted the Earth System Model simulations that provided the ocean physical fields to drive the ecosystem model. C.B. provided expertise on, and code for, the statistical calculations.

## Additional information

**Competing interests:** The authors declare no competing interests.

