## [Peer Review File · Nature Communications]

Reviewers' comments:

Reviewer #1 (Remarks to the Author):

Review of on "Ocean Colour Signature of Climate Change" by Stephanie Dutkiewicz, Anna E. Hickman, Oliver Jahn, Stephanie Henson, Claudie Beaulieu, Erwan Monier

This manuscript explores a global coupled biogeochemical-ocean physics which includes a optical properties module for signatures of climate change by optical water constituents (such as overall phytoplankton (indicated as Chl-a), phytoplankton composition, CDOM and non-algal particles) versus ocean color itself. It analyses the outcome of model runs from 1995 up to 2100 using for the future time realistic changes in the physical ocean in relationship to predictions of anthropogenic emissions according to Sokolev et al. 2009.

The manuscript claims that numerical models provide not only a tool to explore potential future changes in phytoplankton due to anthropogenic driven climate change; it shows that in addition to that also the ocean color, in its representation as remote sensing reflectance, time series data can be explored and that there certain spectral ranges (475 to 525nm) ranges how a much stronger and earlier change than the overall biomass of phytoplankton (indicated by Chl-a) or other optical water constituents. It claims that the numerical model results also show clearly that an early and strong change in phytoplankton community structure is associated with this change / alterations in ocean color of these spectral ranges. The set-up of such a complex model integrating coupling of ocean physics and ecosystem/biogeochemistry which also includes phytoplankton functional type and a coupling to spectral optical properties has been presented only so far by very few groups worldwide. The specific model used in this study has been well presented, described and evaluated in several publications before. The usage of this model in this manuscript to explore climate change not only for oceanic biogeochemical/ecosystem components but also for ocean color is totally novel. The purpose is that only ocean color data are the only (near-)global coverage observation which relate to surface biogeochemistry and it is necessary to interpret this information correctly because by applying algorithms to these data global information on surface biogeochemistry is derived. These algorithms are either to a high or still to a considerable extent dependent on empirical assumptions based on current observations. With the presented model data set the effect on (un)certainty of "standard" ocean color chl-a algorithms under climate change is analyzed by comparison to the "real (modelled!)" chl-a concentration. In summary, the study shows the potential of such a complex coupled model and its simulations to be used not only to identify signatures of global change but also to improve and test algorithms development of using satellite ocean color to derive surface biogeochemistry components under climate change. So it will be of interest not only to the climate community but also to the ocean color community. However, the current discussion on the spectral signature found in the model is quite coarse and does not show the linkage between the spectral resolution of the current optical module of the model (and current satellite ocean color observation)

versus the one needed to understand the changes. Also the perspective to increase the spectral resolution in the coupled model in future should be discussed more clearly. Mostly the relevant literature is cited appropriately, except for two cases outlined below.

Overall, the manuscript is very well written and explores present day and future runs of a numerical biogeochemical-ocean model in a novel way. It clearly states the hypotheses and the way how the study is investigating these and proving them (to a certain respect). The model data are technically sound and the paper provides strong evidence for nearly all of its conclusions. In addition to the more in depth discussion of spectral aspects, there are a few points which need corrections or clarification before this manuscript can be accepted:

1) Main Text, 3rd sentence: The remote sensing reflectance is not the ratio of the upwelling irradiance to the downwelling irradiance, instead the ratio of the upwelling radiance to the downwelling irradiance is used. This needs to be corrected. The derivation of RRS from model output described in the method part of the MS is correct.

2) Language / Typo:

-Main text, 2nd last paragraph, last sentence: Change to “Put in another way,...”

-Page 12, 3rd paragraph add gap between “SeaWiFS and” and “OC-CCI”

3) Missing references:

- Page 4, last sentence in 2nd paragraph: The citation of Wolanin et al. (2017) should be given here, not the citation to Bracher et al. (2017), because in Wolanin et al. exactly this is analyzed for a global simulated set-up.

- Paragraph 6 on page 6, 3rd sentence: it is claimed that these are wavebands (again which ones) where there is the least interannual variability in real world satellite measurements – but no citation is given here where this has been shown.

- See comment 5:

4) Paragraphs 6 on page 6: As mentioned above ,this discussion needs to be extended and requires much more explanations: a) the exact spectral range of which bands are meant here needs to be provided, e.g. it is not clear which spectral region is meant in the 1st and 2nd sentences of this paragraph – is this 425-475nm or 525 to 575 nm? b) In the 3rd sentence it is not clear why at these wave bands the signal to noise ratio increases.... , c) last sentence: also provide information which spectral range is meant exactly. Overall mind that the spectral resolution and coverage in the model is different than that for current and former ocean color sensors.

5) First sentence page 7 and Discussion page 8 3rd paragraph: Be clearer what spectral range needs to be better resolved and what would be the necessary resolution (e.g. Lee et al. 2007, Werdell et al. 2014,Wolanin et al. 2016 may help here). Also elaborate if more details/accuracy of parametrizations and components used in the optical module will be necessary and what are limiting

an extension of this module in set-up and resolution. Add reference to paper by Gregg and Rousseaux (2017) and if available other similar references.

6) Figures:

- a) Figure 4: Color scale in a) should have more contrasting colors and add the units of the hue angle, in b) why is the color scale inverted?
- b) Figure 5: I suggest to remove from a) phy, from b) cdm and from c) det and put instead to the respective color bars that the ratio of a_{phy}/a_{tot} , a_{CDOM}/a_{tot} and a_{det}/a_{tot} are shown.
- c) Figure 7: I think vertical should be horizontal and horizontal should be vertical?

Reference:

Gregg, W.W., and Rousseaux, C. S. (2017). Simulating space global ocean radiances. *Front. Marine Sci.* 4:60. doi: 10.3389/fmars.2017.00060

Lee, Z.; Carder, K.; Arnone, R.; He, M. Determination of primary spectral bands for remote sensing of aquatic environments. *Sensors* 2007, 7, 3428–3441.

Werdell, P. J., Roesler, C.S., & J. I. Goes, J.I. Discrimination of phytoplankton functional groups using an ocean reflectance inversion model. *Appl. Optics*, 53, 4833-4849 (2014).

Wolanin A., Soppa M. A., Bracher A., (2016) Investigation of spectral band requirements for improving retrievals of phytoplankton functional types. *Remote Sensing* 8: 871; doi:10.3390/rs8100871

Reviewer #2 (Remarks to the Author):

Review for the manuscript submitted entitled: “Ocean Colour Signature of Climate Change” by Dutkiewicz et al. The submitted manuscript considers an interesting topic. The authors claim that the remote sensing reflectance (RRS) is a more representative and holistic measurement than chlorophyll, and enables the earlier detection of climate-driven changes. The reason why they claim this is true, is the fact that the Reflectances include alterations in other optically important constituents (i.e. CDOM, particulate detrital matter, community structure etc.), in contrast to chlorophyll product alone (as the authors claim in the abstract, discussion, conclusions).

Here there is a fundamental issue that I cannot support the paper further. It is very well known, that one of the major issues of satellite derived chlorophyll is the fact that one cannot separate (especially in CASE II optically complex waters) chlorophyll from CDOM, detrital matters etc. In other words, these optical constituents (e.g. CDOM, detrital matters) are part of the chlorophyll signal, and in most cases cannot even be separated. Thus, chl-a from satellites already includes the signal from these optical constituents. And this is one of the main reasons why satellite chl-a is over/underestimated in comparison with (ground-truth) in situ chlorophyll data. The authors use chl-a from satellites (a product that definitely contains all the aforementioned additional optical constituents) to claim that is not detecting the changes early enough in comparison to pure reflectances. And they claim that responsible for this result is the fact that the RRS include alterations in other optically important constituents, which are not included in the Chlorophyll product. I am sorry, but I cannot be of any further support, as I simply do not agree with this concept.

Some parts of the manuscript are relatively well written, but there are parts like the abstract and intro that are not well written. There are a few clumsy sentences. Many figures (figures 1 to 7) are more or less repeat previously published work (as the authors also mention). So perhaps it would be better to put them as a supplement and present only the new results. I would suggest shortening the paper including only the crystal clear new results.

Also in your figure 7 you should add the satellite derived trends of chl-a (even if it goes only up to 2015) and also add some in situ chlorophyll from the continuous plankton recorder as you mention in your conclusions. I would feel more comfortable to see (that at least where concurrent in situ, and satellite timeseries co-occur) they actually agree with your model (at least at the beginning of the trend 2000-2015).

In general, I am afraid that the significant claims that you are narrating here are not really supported by this analysis. One of the examples but not only (see my first comments) is "We show that Chl-a derived from RRS with an algorithm developed for the current day does not capture the correct trends over the course of the 21st century. New algorithm development needs to keep the oceans' altering optical properties in mind." Strong statements and I am not convinced by this analysis.

We thank the reviewers for their comments. We have altered the text in response. Below the reviewer comments are in black, and our responses are in blue. In the main text, we have also highlighted in blue all sections that have had substantial changes.

Reviewer 1:

This manuscript explores a global coupled biogeochemical-ocean physics which includes an optical properties module for signatures of climate change by optical water constituents (such as overall phytoplankton (indicated as Chl-a), phytoplankton composition, CDOM and non-algal particles) versus ocean color itself. It analyses the outcome of model runs from 1995 up to 2100 using for the future time realistic changes in the physical ocean in relationship to predictions of anthropogenic emissions according to Sokolev et al. 2009.

The manuscript claims that numerical models provide not only a tool to explore potential future changes in phytoplankton due to anthropogenic driven climate change; it shows that in addition to that also the ocean color, in its representation as remote sensing reflectance, time series data can be explored and that there certain spectral ranges (475 to 525nm) ranges how a much stronger and earlier change than the overall biomass of phytoplankton (indicated by Chl-a) or other optical water constituents. It claims that the numerical model results also show clearly that an early and strong change in phytoplankton community structure is associated with this change / alterations in ocean color of these spectral ranges. The set-up of such a complex model integrating coupling of ocean physics and ecosystem/biogeochemistry which also includes phytoplankton functional type and a coupling to spectral optical properties has been presented only so far by very few groups worldwide. The specific model used in this study has been well presented, described and evaluated in several publications before. The usage of this model in this manuscript to explore climate change not only for oceanic biogeochemical/ecosystem components but also for ocean color is totally novel. The purpose is that only ocean color data are the only (near-)global coverage observation which relate to surface biogeochemistry and it is necessary to interpret this information correctly because by applying algorithms to these data global information on surface biogeochemistry is derived. These algorithms are either to a high or still to a considerable extent dependent on empirical assumptions based on current observations. With the presented model data set the effect on (un)certainly of “standard” ocean color chl-a algorithms under climate change is analyzed by comparison to the “real (modelled!)” chl-a concentration. In summary, the study shows the potential of such a complex coupled model and its simulations to be used not only to identify signatures of global change but also to improve and test algorithms development of using satellite ocean color to derive surface biogeochemistry components under climate change. So it will be of interest not only to the climate community but also to the ocean color community. However, the current discussion on the spectral signature found in the model is quite coarse and does not show the linkage between the

spectral resolution of the current optical module of the model (and current satellite ocean color observation) versus the one needed to understand the changes. Also the perspective to increase the spectral resolution in the coupled model in future should be discussed more clearly. Mostly the relevant literature is cited appropriately, except for two cases outlined below.

Overall, the manuscript is very well written and explores present day and future runs of a numerical biogeochemical-ocean model in a novel way. It clearly states the hypotheses and the way how the study is investigating these and proving them (to a certain respect). The model data are technically sound and the paper provides strong evidence for nearly all of its conclusions. In addition to the more in depth discussion of spectral aspects, there are a few points which need corrections or clarification before this manuscript can be accepted:

We thank the reviewer for these comments and agree that we have not done an adequate job on discussing the spectral signature in the model, and in linking to existing satellite sensor resolution. We now include several additional paragraphs and two new figure to address these shortcomings. One figure (Fig 6) shows the break down of the components on the absorption and backscattering spectra, which is important to illustrate the spectral discussion. And the other figure (Fig 3) compares relative magnitude of interannual variability between different optically important constituents and RRS. This figure shows the relatively lower variability in the RRS. These two figures help frame the new discussion.

We have also added the suggested references.

1) Main Text, 3rd sentence: The remote sensing reflectance is not the ratio of the upwelling irradiance to the downwelling irradiance, instead the ratio of the upwelling radiance to the downwelling irradiance is used. This needs to be corrected. The derivation of RRS from model output described in the method part of the MS is correct.

Corrected, thank you

2) Language / Typo:

-Main text, 2nd last paragraph, last sentence: Change to "Put in another way,..."

Not sure where you meant this change. This addition did not seem appropriate in the 2nd last paragraph, last sentence? But we have altered this paragraph anyway in this new version, so potentially this not an issue anymore.

-Page 12, 3rd paragraph add gap between "SeaWiFS and" and "OC-CCI"

Done, thank you

3) Missing references:

- Page 4, last sentence in 2nd paragraph: The citation of Wolanin et al. (2017) should be given here, not the citation to Bracher et al. (2017), because in Wolanin et al. exactly this is analyzed for a global simulated set-up.

Agreed, we've changed this.

- Paragraph 6 on page 6, 3rd sentence: it is claimed that these are wavebands (again which ones) where there is the least interannual variability in real world satellite measurements – but no citation is given here where this has been shown.

This was an analysis that we did ourselves, hence the lack of a citation. Though we assume that this must be well known in the ocean colour community, but have not been able to find a references. As such, we realise now that this analysis (see Fig 3) is important to include in the revised manuscript. We have also added Figure 6 which shows that the RRS in the blue-green wavelengths have the lowest relative variability. We discuss these issues in the following section of text:

Pg 3: “The model also captures the observed median magnitude of the Chl-a variability relative to the mean (Fig 3a). The slightly lower variability is expected as we do not capture all the sources of variability (e.g. mesoscale features) found in the real ocean. The current day relative magnitude of interannual variability of the other optically important constituents (CDOM, detrital particles) are predicted by the model to have similar values to Chl-a. However, the relative magnitude of the observed interannual variability of R_{RS} is much lower (red symbols in Fig 3b) at the lower wavebands. The model captures these observational values over most of the spectrum, but not at the high wavebands (Fig 3b). However, we note that the uncertainty in R_{RS} is higher at higher wavebands²¹.”

Pg 6: “These are the wavebands where there is the least interannual variability in the model (Fig 3b). This mirrors the observed dip in the relative magnitude of the interannual variability in the 490nm and 510nm wavebands of the OC-CCI products. “

- See comment 5:

4) Paragraphs 6 on page 6: As mentioned above ,this discussion needs to be extended and requires much more explanations: a) the exact spectral range of which bands are meant here needs to be provided, e.g. it is not clear which spectral region is meant in the 1st and 2nd sentences of this paragraph – is this 425-475nm or 525 to 575 nm? b) In the 3rd sentence it is not clear why at these wave bands the signal to noise ratio increases.... , c) last sentence: also provide information which spectral range is meant exactly. Overall mind that the spectral resolution and coverage in the model is different than that for current and former ocean color sensors.

Agreed. We have now added more detail, and significantly altered this paragraph.

- a) The two wavebands which show the strongest signal are centered at 475nm and 500nm (in our model the bands are 25nm wide). We now make the specific values more explicit, and note the coarseness of the wavebands

Pg 6: "The strongest signal is in the blue-green range (in our model the two 25nm wavebands centred at 475 and 500nm); these wavebands show a clear signal over 50% of the ocean by the end of the century. The strongest signal of trend was found for over the wavelengths spanning from 487-512nm, with 63% of the ocean providing a significant signal by 2100."

- b) Signal to noise was potentially misleading wording to have used, we are now clear that we mean "relative magnitude of interannual variability". We point to the text on Pg 3 cited above in comment (3). And also the following text:

Pg 6: "These are the wavebands where there is the least interannual variability in the model (Fig 3b). This mirrors the observed dip in the relative magnitude of the interannual variability in the 490nm and 510nm wavebands of the OC-CCI products."

- c) We make much clearer statements about the exact wavebands we consider:

Pg 6: "The strongest signal is in the blue-green range (in our model the two 25nm wavebands centred at 475 and 500nm)"

Pg 8: "Our work identifies 467-512nm as promising from this aspect. Unfortunately, the model bands are too coarse (at 25nm) to provide more details of the best wavelength and bandwidth most suitable for trend analysis."

We also add some sentences that link more closely to the satellite sensors here and in more depth in the discussion. This is an important part of the discussion and thank the reviewer for suggesting it.

Pg 6: "The low variability together with sensitivity to changes in all water constituents suggest that these wavebands in the real world satellite sensors (490nm and 510nm) might be the first measurements to detect climate change signal in the marine ecosystems. "

Pg 8: "There is considerable effort expended to determine the best spectral bands for satellite sensors^{36,41,}, and numerical models are starting to be used to explore aspects of bands for future mission (e.g. PACE)³⁸. We suggest that the choice of bandwidths for future satellite sensors should also include the strength of trends they will capture. Our work identifies 467-512nm as promising from this aspect. However, the current model bands are too coarse (at 25nm) to provide more details of the best wavelength and bandwidth most suitable for trend analysis. However, current and historic sensors (e.g. SeaWiFS, MODIS, VIIRS, MERIS) have all included wavebands around 490nm and we suggest that it is

imperative to maintain a band that is compatible in future missions for the earliest signatures of marine ecosystem changes.”

Pg 9: “Our results thus suggest several focus areas important for monitoring the response of ocean productivity to climate change: 1) maintaining ocean colour sensors compatibility and long term stability, particularly in the 490nm waveband;”

5) First sentence page 7 and Discussion page 8 3rd paragraph: Be clearer what spectral range needs to be better resolved and what would be the necessary resolution (e.g. Lee et al. 2007, Werdell et al. 2014, Wolanin et al. 2016 may help here).

We have significantly re-written this preceding paragraph to make the ranges clearer. See above quoted text (Comment 4 and 5). The link to the papers you mention (which we now reference) was very helpful.

Also elaborate if more details/accuracy of parametrizations and components used in the optical module will be necessary and what are limiting an extension of this module in set-up and resolution. Add reference to paper by Gregg and Rousseaux (2017) and if available other similar references.

We believe this belongs in the discussion rather than at this location, but have added a new section, and include the following text:

Pg 7: “Though relatively complex, the model still has only a limited number of optically different plankton species and does not include several important optical constituents (e.g. viruses, mineral) or the effects of salinity¹⁸ that could impact the accuracy of the reflectance and also the ability to capture the natural interannual variability.”

Pg 8 “The model has relative coarse (25nm) wavebands, which differ in size and spacing from historic and current satellite sensors. Thus the model can only provide broad estimates of the wavelengths, and cannot in its current form suggest bandwidths, that will provide the best long term change signals. However, given these caveats, the model still provides a unique opportunity to estimate the ocean colour changes over the 21st century, and what signals we should plan to explore most to detect changes.”

In the discussion we add reference to Gregg and Rousseaux (for links to PACE) as well as other similar references (though note that only 3 other large scale models have the capabilities we show here, and none have considered climate change).

Pg 8: “Very few models currently include a radiative transfer component or produce products such as reflectance^{16,38,39,40}, and no model with these capabilities has looked at a climate change scenario.”

6) Figures:

a) Figure 4: Color scale in a) should have more contrasting colors and add the units of the hue angle,

Done. We had tried to be closer to “real” colours, but realize this fails to provide adequate visual range, so have stretched the colour scale.

in b) why is the color scale inversed?

Decreasing values lead to redder part of spectrum, and increasing to bluer. So we thought that having the colour bar follow this pattern made sense. But we see this could be confusing relative to all the other difference plots. So we now switch back to the more traditional colour bar.

b) Figure 5: I suggest to remove from a) phy, from b) cdm and from c) det and put instead to the respective color bars that the ratio of a_{phy}/a_{tot} , a_{CDOM}/a_{tot} and a_{det}/a_{tot} are shown.

We assume you meant Figure 6, not 5. We have changed as suggested, though note this is now Fig 7.

c) Figure 7: I think vertical should be horizontal and horizontal should be vertical?

Indeed. Thank you for catching that.

Reference:

Gregg, W.W., and Rousseaux, C. S. (2017). Simulating pace global ocean radiances. *Front. Marine Sci.* 4:60. doi: 10.3389/fmars.2017.00060

Lee, Z.; Carder, K.; Arnone, R.; He, M. Determination of primary spectral bands for remote sensing of aquatic environments. *Sensors* 2007, 7, 3428–3441.

Werdell, P. J., Roesler, C.S., & J. I. Goes, J.I. Discrimination of phytoplankton functional groups

using an ocean reflectance inversion model. *Appl. Optics*, 53, 4833-4849 (2014).

Wolanin A., Soppa M. A., Bracher A., (2016) Investigation of spectral band

requirements for improving retrievals of phytoplankton functional types. Remote Sensing 8: 871; doi:10.3390/rs8100871

We now include all these references. Thank you for suggesting them.

Reviewer #2 (Remarks to the Author):

The submitted manuscript considers an interesting topic. The authors claim that the remote sensing reflectance (RRS) is a more representative and holistic measurement than chlorophyll, and enables the earlier detection of climate-driven changes. The reason why they claim this is true, is the fact that the Reflectances include alterations in other optically important constituents (i.e. CDOM, particulate detrital matter, community structure etc.), in contrast to chlorophyll product alone (as the authors claim in the abstract, discussion, conclusions).

Here there is a fundamental issue that I cannot support the paper further. It is very well known, that one of the major issues of satellite derived chlorophyll is the fact that one cannot separate (especially in CASE II optically complex waters) chlorophyll from CDOM, detrital matters etc. In other words, these optical constituents (e.g. CDOM, detrital matters) are part of the chlorophyll signal, and in most cases cannot even be separated. Thus, chl-a from satellites already includes the signal from these optical constituents. And this is one of the main reasons why satellite chl-a is over/underestimated in comparison with (ground-truth) in situ chlorophyll data. The authors use chl-a from satellites (a product that definitely contains all the aforementioned additional optical constituents) to claim that is not detecting the changes early enough in comparison to pure reflectances. And they claim that responsible for this result is the fact that the RRS include alterations in other optically important constituents, which are not included in the Chlorophyll product. I am sorry, but I cannot be of any further support, as I simply do not agree

with this concept.

We are very confused by the above paragraph. For much of the paper we consider how the model “actual” optically important constituents change over the 21st century. As explicitly stated in the paper, “model actual” is comparable to in situ measurements. Chl trends in Fig 8 and black line in Fig 7 are from “model actual” Chl. This is not a derived product that has “major issues”. We do not “use chl-a from satellites (a product that definitely contains all the aforementioned additional optical constituents) to claim that is not detecting the changes early enough in comparison to pure reflectances” (as the reviewer claims above). In fact, later in the paper we do look at the “model derived” Chl-a product (red line in Fig 7) and make the same statement as the reviewer, that other “optical constituents (e.g. CDOM, detrital matters) are part of the chlorophyll signal, and in most cases cannot even be separated”, where “chlorophyll” now is the satellite-like derived product. We then go on to make the statement that derived Chl product can’t capture the trends as it also captures the changes to the other optical constituents. We find that this point is made several times in the course of the paper, and are confused as to how the reviewer missed this.

The fact we can separately look at “actual” and “derived” Chl from our model is unique and major step forward in linking numerical models and ocean colour. We, in particular, make a major point about the difference between the “actual” and “derived” product (the latter which has “major issues”).

To assure that other readers do not make the same assumptions, we now add clarification of “actual” and “derived” Chl-a in the last paragraph of the introduction, alter the abstract to include “in water Chl-a”, and bring the discussion of how we calculate the “derived” Chl-a into the first section “The Present Day”. Figure 1 now shows both model actual and model derived Chl-a so the distinction is made pictorially too.

That changes to all the constituents integrate to change the reflectance is a fact, and a main motivation for measuring ocean colour. Thus we are unsure why the reviewer does not like this statement. We have shown in Fig 6 of the old text (and even more so with the newer version of the figure – now Fig 7) how the different constituents have differing roles in the absorption and scattering.

Some parts of the manuscript are relatively well written, but there are parts like the abstract and intro that are not well written. There are a few clumsy sentences. Many figures (figures 1 to 7) are more or less repeat previously published work (as the authors also mention). So perhaps it would be better to put them as a supplement and present only the new results. I would suggest shortening the paper including only the crystal clear new results.

It is important to show the model basic results to set the context and especially important for a more general audience who might not know what a typical Chl-a/biomass/RRS distribution might be. Due to space constraints we have now moved Fig 2 to the supplemental. But, we elect to keep figures 1 and 3 (now 1 and 2) in the main text. We also now use Fig 1 to demonstrate the difference between model “actual” and model “derived” Chl-a early in the paper, and include standard deviations. These are new and important.

Otherwise all additional figures are new and important to include. We believe we have only included the newest and most exciting results here.

Also in your figure 7 you should add the satellite derived trends of chl-a (even if it goes only up to 2015) and also add some in situ chlorophyll from the continuous plankton recorder as you mention in your conclusions. I would feel more comfortable to see (that at least where concurrent in situ, and satellite timeseries co-occur) they actually agree with your model (at least at the beginning of the trend 2000-2015).

The model is an earth system model with its own internal variability, therefore the interannual variability (e.g. El Nino) do not occur at the same time (though do in frequency) as the real world. Thus to plot the timeseries would be confusing. CPR does not provide Chl data, and we do not believe there is a sensible way to include a timeseries of in situ Chl here, and certainly beyond the scope of this paper. We also point out that our results suggest that we are unlikely to see a trend in the 1998-2015 timeframe, and thus feel it would be incorrect to even include the trends found in the OC-CCI data. We do however now include a discussion about how the model captures the natural variability and include additional panels on Fig 1:

Pg 3: “The ocean ecosystems and ocean colour are not static, changing with the seasons and with inter-annual climate variability (Fig 1d,e,f). The model (Fig 1e) captures the patterns of the satellite

estimated variability (Fig 1f), but under-estimates in the subtropical gyres and over-estimates in the higher latitudes (where it also overestimates the mean). The model also captures the observed median magnitude of the Chl-a variability relative to the mean (Fig 3a). The slightly lower variability is expected as we do not capture all the sources of variability (e.g. mesoscale features) found in the real ocean. The current day relative magnitude of interannual variability of the other optically important constituents (CDOM, detrital particles) are predicted by the model to have similar values to Chl-a.”

In general, I am afraid that the significant claims that you are narrating here are not really supported by this analysis. One of the examples but not only (see my first comments) is “We show that Chl-a derived from RRS with an algorithm developed for the current day does not capture the correct trends over the course of the 21st century. New algorithm development needs to keep the oceans’ altering optical properties in mind.” Strong statements and I am not convinced by this analysis.

We are surprised that the author finds this statement “strong”. When we have presented this work at ocean colour conferences, the audience has strongly agreed with this statement. Indeed, it is operational practice to update algorithms over time and this recognised need motivates significant ongoing in situ sampling. We have justified this point using our model and make the point in the particular context of climate. We also disagree that we have not supported this point. Our analysis shows that model absorption changes over the course of the 21st century (see Fig 7). Our model is self-consistent: changes of the IOPs relative to one another change the optical make-up of the water; algorithms developed for today’s ocean will not work as well in a future ocean with different optics. By actually going through the process of deriving Chl from RRS and looking at the trends we have shown that the derived Chl-a does not match the actual Chl-a trend. We feel that our analysis is definitely adequate to prove this point. And thus within the model framework we have adequately shown that the ocean experiences “altering optical properties” and that the standard Chl algorithm does not continue to work as well later in the century.

The reviewer could have instead taken issue that the optical properties might not change in the real ocean as suggested in our model. We have now added sentences to highlight the missing processes in our model, and provide a caveat that our model has limitations in capturing the changes in optical properties.

Pg 7/8: “These results presented in this manuscript must be taken in the context of the simplifications necessary in the ocean ecosystem and optical model. Though relatively complex, the model still has only a limited number of optically different plankton species and does not include several important optical constituents (e.g. viruses, mineral) or the effects of salinity¹⁸ that could impact the accuracy of the

reflectance and also the ability to capture the natural interannual variability. The parameterization of CDOM and detrital matter is still simplistic, and we caution that the model can not yet be used to make strong definitive comments on these changes, but can provide unique chance to explore the potential changes.”

Reviewers' comments:

Reviewer #1 (Remarks to the Author):

I am happy with the revision and think the authors have responded very well and improved a lot the manuscript in respect to my former comments. I have just a minor point which would be valid to be corrected:

1) Line 271ff. I think here it is good to cite the new algorithm developments in terms of considering different water types, a good reference for that is: Hieronymi, M., Müller, D., Doerffer, R. The OLCI Neural Network Swarm (ONNS): A Bio-Geo-Optical Algorithm for Open Ocean and Coastal Waters. *Front. Mar. Sci.*, 11 May 2017.

So after that I think the paper can be accepted for publication in Nature Communications.

Reviewer #3 (Remarks to the Author):

I am a new reviewer of this manuscript, so this revised version is the first time I've had an opportunity to evaluate the work. My overall conclusion is that this is an interesting study that provides some new insights on potential ocean ecosystem signals of anthropogenic climate change. The manuscript is also very well written and appropriately so for a broader audience. Well done! I have also evaluated the authors' responses to the previous 2 reviewers' comments and found that the changes made to the manuscript were substantial and appropriate. I found that comments from reviewer #2 were largely in error and that the authors responses were appropriate. My conclusion is that the manuscript is suitable for publication in Nature Communications, although I have a few of my own comments that I ask the authors to consider.

1. In the abstract and first paragraph of the main text, the authors state that satellites do not directly measure chlorophyll, but rather that chlorophyll is a derived product of the directly measured RRS. Strictly speaking, RRS is also not directly measured by satellites. What satellites measure directly is top of atmosphere radiances and RRS is derived from these data by applying an atmospheric correction (which can include substantial errors). This is a minor detail, but should be clarified.

2. In multiple places in the current manuscript it is suggested that changes in surface chlorophyll result from changes in nutrient availability (which the authors link to increased surface stratification). While there is a first-order relationship between global chlorophyll distributions and nutrients, when evaluating anomalies (a much smaller signal) at a given location a link between chlorophyll changes and nutrients may not hold (and in fact, in oligotrophic regions should not be assumed). I suggest the authors take a look at:

Lozier, M. S., Dave, A. C., Palter, J. B., Gerber, L. M. & Barber, R. T. On the relationship between stratification and primary productivity in the North Atlantic. *Geophys. Res. Lett.* 38, L18609 (2011).

What that paper shows is that changes in SST or stratification at an oligotrophic site have little impact on nutrient budgets because the surface mixed layer is far removed from the nutricline. This point was developed at a global level in the following paper:

Behrenfeld, M. J., R. T. O'Malley, E. S. Boss, T. K. Westberry, J. R. Graff, K. H. Halsey, A. J. Milligan, D. A. Siegel, and M. B. Brown (2016), Reevaluating ocean warming impacts on global phytoplankton, *Nat. Clim. Change*, 6, 3223–3330.

What that paper shows is that satellite detected anomalies in surface chlorophyll over most of the global ocean are largely reflective of photoacclimation responses to changing mixed layer light levels. More simply, if the surface mixed layer shoals, a photoacclimation response is guaranteed but an impact on nutrient availability is less likely. I noticed in the Methods section of the current paper that the model includes a photoacclimation component. Thus model results on future chlorophyll changes reflect both light and nutrient effects. As the current analysis does not distinguish these two drivers of chlorophyll change, both factors should be acknowledged as responsible for trends in chlorophyll

3. The current analysis is very much limited to traditional satellite chlorophyll algorithms based on simple wavelength ratios. While the conclusions drawn are appropriate for such algorithms, they may not be appropriate for inversion algorithms (e.g., QAA, GSM, GIOP). Ideally, these newer algorithms separate contributions from cDOM, phytoplankton pigments, and backscattering and thus may not (again, ideally) be compromised by future changes in the relative contributions from cDOM, backscattering, chlorophyll, and community composition. With respect to the PACE mission mentioned on line 316, it is noteworthy that this particular mission will include retrievals in the near UV (which will improve separation of cDOM and pigment absorption) and hyperspectral measurements through the visible which should improve the detection of changing community composition (such as predicted by the current model). Perhaps some additional statements in the manuscript would be appropriate....

4. In multiple places in the manuscript (one example being lines 284-285), it is suggested that the model is performing well because it reproduces the patterns and magnitude of satellite-observed chlorophyll and/or RRS patterns. It is also stated that the model captures observed interannual variability in these properties. Is this true? Evidence for these statements is given in figures 1 and 3 and it is up to the reader to determine if the model and observations agree. With respect to figure 1, in my eyes at least, there is some level of agreement but also considerable differences. The critical issue here is that this is a comparison of first-order global patterns in chlorophyll and the focus of the manuscript is on interannual anomalies. If the model fails to capture significant first-order patterns, how much confidence can the reader have in modeled anomalies, which are much much smaller? Some more rigorous statistical assessments of model performance would be helpful. With respect to figure 3, the comparison being made here is for globally-integrated values of interannual variability. This level of spatial integration undoubtedly greatly dampens the level of variability occurring at much smaller scales and may also misrepresent the agreement between model and measured data at these smaller scales. Since many of the interesting results of the current study revolve around spatially-resolved patterns, it is not clear that a globally-integrated comparison of measured/modeled properties is appropriate. Again, statistical evidence of model performance at the spatial scale of interpreted results would be helpful.

5. Here's an easy one... On line 288, I think 'relative' should be 'relatively' :)

I hope these comments are helpful. Again, nice job!

We thank the reviewers for taking the time to evaluate our manuscript and for their positive and useful feedback. We believe that these comments have helped make this article clearer and more relevant. Below the reviewers' comments are in black and our responses in blue. In the revised manuscript, changes to the text that address these comments are also in blue; there are other minor alterations to the text to remain below the word limit.

Reviewers' comments:

Reviewer #1 (Remarks to the Author):

I am happy with the revision and think the authors have responded very well and improved a lot the manuscript in respect to my former comments. I have just a minor point which would be valid to be corrected:

1) Line 271ff. I think here it is good to cite the new algorithm developments in terms of considering different water types, a good reference for that is: Hieronymi, M., Müller, D., Doerffer, R. The OLCI Neural Network Swarm (ONNS): A Bio-Geo-Optical Algorithm for Open Ocean and Coastal Waters. *Front. Mar. Sci.*, 11 May 2017.

So after that I think the paper can be accepted for publication in Nature Communications.

We thank the reviewer for this and the previous review. We agree that Hieronymi et al (2017) is an excellent paper to include. It is reference 42 and is cited in line 277 and 347 of the revised text.

Reviewer #3 (Remarks to the Author):

I am a new reviewer of this manuscript, so this revised version is the first time I've had an opportunity to evaluate the work. My overall conclusion is that this is an interesting study that provides some new insights on potential ocean ecosystem signals of anthropogenic climate change. The manuscript is also very well written and appropriately so for a broader audience. Well done! I have also evaluated the authors' responses to the previous 2 reviewers' comments and found that the changes made to the manuscript were substantial and appropriate. I found that comments from reviewer #2 were largely in error and that the authors responses were appropriate. My conclusion is that the manuscript is suitable for publication in Nature Communications, although I have a few of my own comments that I ask the authors to consider.

We thank the reviewer very much for providing a positive assessment of our work and for taking the time to carefully evaluate our manuscript and the previous reviews as well.

1. In the abstract and first paragraph of the main text, the authors state that satellites do not directly

measure chlorophyll, but rather that chlorophyll is a derived product of the directly measured RRS. Strictly speaking, RRS is also not directly measured by satellites. What satellites measure directly is top of atmosphere radiances and RRS is derived from these data by applying an atmospheric correction (which can include substantial errors). This is a minor detail, but should be clarified.

We agree that this was not strictly speaking correct. Given the word restriction we can only allude to this in the abstract, and now have changed to:

Lines 17-19: "However, satellite sensors do not measure Chl-a directly. Instead, Chl-a is estimated from remote sensing reflectance (R_{RS}): the ratio of upwelling radiance to the downwelling irradiance at the ocean's surface.

To clarify this point more clearly, we have altered the introduction:

Lines 31-35: "Ocean colour satellite sensors measure the radiance at the top of atmosphere over a range of wavelengths. After taking account of the optically significant constituents in the atmosphere (which can include a substantial error)¹, a key product of ocean colour is remotely sensed reflectance (R_{RS}), the ratio of the upwelling radiance to the downwelling irradiance at the ocean surface. R_{RS} is the standard product provided by space agencies.

2. In multiple places in the current manuscript it is suggested that changes in surface chlorophyll result from changes in nutrient availability (which the authors link to increased surface stratification). While there is a first-order relationship between global chlorophyll distributions and nutrients, when evaluating anomalies (a much smaller signal) at a given location a link between chlorophyll changes and nutrients may not hold (and in fact, in oligotrophic regions should not be assumed). I suggest the authors take a look at:

Lozier, M. S., Dave, A. C., Palter, J. B., Gerber, L. M. & Barber, R. T. On the relationship between stratification and primary productivity in the North Atlantic. *Geophys. Res. Lett.* 38, L18609 (2011).

What that paper shows is that changes in SST or stratification at an oligotrophic site have little impact on nutrient budgets because the surface mixed layer is far removed from the nutricline. This point was developed at a global level in the following paper:

Behrenfeld, M. J., R. T. O'Malley, E. S. Boss, T. K. Westberry, J. R. Graff, K. H. Halsey, A. J. Milligan, D. A. Siegel, and M. B. Brown (2016), Reevaluating ocean warming impacts on global phytoplankton, *Nat. Clim. Change*, 6, 3223–3330.

What that paper shows is that satellite detected anomalies in surface chlorophyll over most of the global ocean are largely reflective of photoacclimation responses to changing mixed layer light levels. More simply, if the surface mixed layer shoals, a photoacclimation response is guaranteed but an impact on nutrient availability is less likely. I noticed in the Methods section of the current paper that the model includes a photoacclimation component. Thus model results on future chlorophyll changes reflect both light and nutrient effects. As the current analysis does not distinguish these two drivers of chlorophyll

change, both factors should be acknowledged as responsible for trends in chlorophyll

The model Chl-a changes (Fig 4a) have very similar patterns to those for primary production, suggesting that in the long term, the reduced supply nutrients may be a stronger effect than photoacclimation. But we do note that the patterns are not completely correlated and that processes such as mentioned by the reviewer have an influence that may be important in some regions. In the model there are indeed interesting patterns of change in Chl:C, which are driven in part by changes in stratification, but also changes in nutrient stress and temperature (which in the Geider formulation we use also affects Chl:C). We find that one of the strongest drivers in changes in Chl:C in our model is the shift from diatoms to other phytoplankton, since the model parameterizes diatoms to have a higher maximum Chl:C (as indicated by culture studies, e.g. Geider et al. 1997 MEPS 148:187-200). We are interested to explore the changes in Chl-a:C more, but this is not within the scope of this paper. However, as suggested by the reviewer, we introduce these ideas in the following.

Firstly, we try to emphasize that it is not only stratification that alters nutrient supplies, in fact we believe that the alterations to circulation are more important (though this is not an issue to address in this paper, but does link to the finding of Lozier et al, 2011):

Line 117-120: *“Over the course of the 21st century mean global sea surface temperature (SST) increases by 3°C, there is increased stratification and reduced mixing at the surface. The meridional overturning circulation slows and shallows relative to current day conditions. These changes lead to a reduction in the supply of macronutrients from depth.”*

Next, we include physiology as an additional driver of Chl-a changes:

Line 122: *“The shifts in Chl-a (Fig 4a) reflect multiple physical and physiological changes.”*

And more explicitly mention Chl:C changes (reference 26 is Behrenfeld et al 2016):

Line 124-130: *“In other mid to high latitude regions a complex combination of stratification-induced reduction of light limitation (impacting both growth rates, and photo-acclimation²⁶), decreased nutrient supply, and increased growth rates due to warmer temperatures, leads to a mixed pattern of positive and negative responses¹³. In general, Chl-a changes in the same direction as primary production, but subtle differences suggest that alterations in Chl:C ratios also play a role. In the model Chl:C ratios are driven by light, temperature, and nutrient stress²⁷.”*

Lines 176-179: *“Chl:C ratios are also altered by such changes in the community structure. Thus, in different regions of the ocean various combinations of relative changes occur depending on the local alterations to stratification, productivity, community structure and photo-acclimation, driving differing effects on reflectance.”*

3. The current analysis is very much limited to traditional satellite chlorophyll algorithms based on simple wavelength ratios. While the conclusions drawn are appropriate for such algorithms, they may not be appropriate for inversion algorithms (e.g., QAA, GSM, GIOP). Ideally, these newer algorithms separate contributions from cDOM, phytoplankton pigments, and backscattering and thus may not

(again, ideally) be compromised by future changes in the relative contributions from cDOM, backscattering, chlorophyll, and community composition.

Yes, we agree that this should be addressed further. We had included a brief mention of such inversion techniques in the introduction (line 35-37). Here now we add an additional reference to the inversion techniques (Werdell et al 2018, reference 4). We further discuss these other types of methods in several places, including speculating that (though out of the scope of this paper), a similar model system might be useful to consider whether semi-inversion techniques can assist in more rapid or robust trend detection. See changes:

Lines 281-284: *“On the other hand, there are also developments in semi-empirical inversion algorithms^{3,4} that also estimate the contributions of CDOM, particle backscattering, as well as Chl-a. As such, these techniques may be less affected by the changing relative importance of the different water constituents.”*

Lines 346-347: *“Techniques that separately estimate the different water constituents^{4,42} may provide a better avenue for detecting trends.”*

Lines 467-470: *“However, there are other approaches (e.g. semi-analytical inversion)^{3,4} that attempts to more mechanistically estimate not only Chl-a concentration, but also other constituents such as CDOM. We believe that exploring whether this approach will allow climate change trends to be more rapidly or robustly detected will be a promising avenue for future study.”*

With respect to the PACE mission mentioned on line 316, it is noteworthy that this particular mission will include retrievals in the near UV (which will improve separation of cDOM and pigment absorption) and hyperspectral measurements through the visible which should improve the detection of changing community composition (such as predicted by the current model). Perhaps some additional statements in the manuscript would be appropriate....

This is a good point, and I think our manuscript provides additional rational for including these bands in future missions. We added to the discussion:

Lines 354-356: *“Our results also suggest that including sufficient bands to detect different communities (e.g. hyperspectral) and those that will help separate signals of CDOM (e.g. ultra-violet) will be important for monitoring ecosystem changes (as is planned for the PACE mission).”*

4. In multiple places in the manuscript (one example being lines 284-285), it is suggested that the model is performing well because it reproduces the patterns and magnitude of satellite-observed chlorophyll and/or RRS patterns. It is also stated that the model captures observed interannual variability in these properties. Is this true? Evidence for these statements is given in figures 1 and 3 and it is up to the reader to determine if the model and observations agree. With respect to figure 1, in my eyes at least, there is some level of agreement but also considerable differences. The critical issue here is that this is a comparison of first-order global patterns in chlorophyll and the focus of the manuscript is on interannual anomalies. If the model fails to capture significant first-order patterns, how much

confidence can the reader have in modeled anomalies, which are much much smaller? Some more rigorous statistical assessments of model performance would be helpful.

We now include more rigorous statistics, and agree that the paper needs more description of the discrepancies. We include 4 new supplemental figures (Taylor diagram, and plots of biases), and have altered the main text to describe some of the good/not so good comparisons, and a substantial addition to the methods on this subject.

In Lines 66-97 we reference the new Supplemental Figures (S1-S4), and add the following text for describing discrepancies:

Lines 97-105: *“The model (Fig 1e) in general captures the patterns of the satellite estimated variability (Fig 1f), though over-estimates the interannual variability in higher latitudes, where it also overestimates the mean (Supplemental Fig S2). The model Equatorial Pacific has a narrower band of variability around the equator than seen in the observations (Supplemental Fig S3), a discrepancy that shows up in both Chl-a and R_{RS} . The R_{RS} variability is otherwise generally underestimated. We find it instructive to also examine the magnitude of the interannual variability relative to the climatological mean composite (Fig 3, Supplemental Fig S4). The model’s slightly lower value relative to the observed median magnitude of this ratio in Chl-a (Fig 3a) is expected as the model does not capture all the sources of variability (e.g. mesoscale features) found in the real ocean.”*

Lines 295-297: *“However, there are discrepancies, with the model overestimating Chl-a in high latitudes, and suggesting a much more limited region of variability in the Equatorial Pacific than observed.”*

Methods:

Lines 475-487: *“We compare mean composites (and the interannual variability of these composites) over 1998 to 2015. Composites are derived from all monthly means where there are satellite measurements. Thus for instance high latitudes only have input for months where there is sufficient light, and some equatorial regions miss months when there are too many clouds. Model composites are derived with the same missing months to match the observations. The model internal interannual variability does not match the real world (i.e. El Ninos do not occur in the same years), thus we compare interannual variability in terms of a temporal standard deviation of annual composites from 1998 to 2015 (Fig 1, Supplemental Fig S3). To evaluate the skill of the model, we compare to satellite observations, which constitute only an estimate of “true” Chl-a, and R_{RS} with potential uncertainties due to atmospheric corrections¹. It must also be noted that the potential presence of discontinuities due to merging measurements from different sensors in the satellite record may also bias comparison with the model⁶⁵. Indeed, the model does not contain such discontinuities, and measures of agreement between the model and observations such as correlation and relative bias (Supplemental Fig S1-S4) may underestimate how the model captures central tendency and interannual variability in the observations.”*

Line 492-495: *“The model has Chl-a too high relative to the OC-CCI product in high latitudes (sometimes by a factor of 2 or more), and there are some patches of the subtropical gyres that are too low. Regions of the equatorial Atlantic and Indian oceans are too high in the model. The region of high Chl-a in the equatorial Pacific is narrower in the model than the observations.”*

Lines 507-521: *“We further evaluate the model R_{RS} (Fig 2, Supplemental Fig S1, S2, S3, S4) in a similar manner to Chl-a. The model R_{RS} are linearly interpolated from the 25nm bands to the same bands as the OC-CCI product. The model captures the reversed patterns between blue (412,443nm) and green (555nm) R_{RS} between gyres and highly productive regions. Several of the model biases reflect the biases seen in the Chl-a: underestimation of the blue band in the subtropics where modelled Chl-a is too low relative to the real ocean, and higher green R_{RS} in the equatorial Atlantic and Indian Ocean than the satellite measurements where the model overestimates Chl-a relative to the satellite product (Fig 1, Supplemental Fig 2). Additionally, the wavebands all show discrepancies in the Equatorial Pacific where the modelled Chl-a is too narrowly confined to the equator. The effect of salinity on R_{RS} is not captured in the model, likely leading to additional discrepancies especially in the low latitudes where effects of salinity become more important¹⁸. Some equatorial regions (especially the Atlantic) also have high cloud cover, and in such regions the satellite product may be biased. Correlations between model and OC-CCI (Supplemental Fig S1) are best for the low and high wavebands, and worst for the blue-green (490 and 510). These latter are the wavebands where the reversal of the patterns of high/low R_{RS} occur (see Supplemental Fig S2) and are thus most difficult to capture correctly. For the same reason, these are the wavebands where the linear interpolation from the model bands to those of the OC-CCI products are most problematic.”*

With respect to figure 3, the comparison being made here is for globally-integrated values of interannual variability. This level of spatial integration undoubtedly greatly dampens the level of variability occurring at much smaller scales and may also misrepresent the agreement between model and measured data at these smaller scales. Since many of the interesting results of the current study revolve around spatially-resolved patterns, it is not clear that a globally-integrated comparison of measured/modeled properties is appropriate. Again, statistical evidence of model performance at the spatial scale of interpreted results would be helpful.

We now include a Supplemental Figure S4 which shows this ratio for several RRS wavebands and Chl-a over the globe. In the methods we discuss the comparisons. We believe that Supplemental Fig S4 is compelling, but decided to maintain the global integrated figure in the main text, feeling that a 21 panel plot is maybe too much. However, we now reference Supplemental Fig S4 in the caption and the text so that an interested party can easily find this figure. Several of the text list above addresses the interannual variability (the temporal STD of the annual composites). We include additional text specifically on the interannual variability and on the ratio of the interannual variability to the mean:

Lines 108-110: *“The model captures these lower ratios (though underestimates the variability, as in Chl-a) over most of the spectrum, but not at the high wavelength bands (Fig 3b, Supplemental Fig S4).”*

Methods:

Lines 489-506: *“The model also captures the patterns of interannual variability (Fig 1e,f, Supplemental Fig S3), but does overestimate it in regions where it also overestimates the 18 year mean composite. Particularly noticeable is that the high variability in the equatorial Pacific is shifted relative to the observations, suggesting the physical manifestation of the El Nino/La Nina response is slightly misplaced in the Earth System Model (unsurprisingly given the too narrow upwelling band as seen in the Chl-a composite). The model also does a good job at capturing the patterns of the ratio of the interannual variability to the 18 year composite (Supplemental Fig S4), though overestimates in the Equatorial Pacific (where there is a mismatch in the physical manifestation of El Nino/La Nina), but otherwise has a low*

bias elsewhere. This latter is expected as we do not capture all the sources of variability (e.g. mesoscale features) found in the real ocean."

Lines 521-547: "In general, the model has a low bias (Supplemental Fig S3) for the interannual variability in R_{RS} , with the most noticeable exception in the equatorial Pacific where the Chl-a mismatch bias occurs. The model also captures the patterns of the ratio of the interannual variability to the 18 year mean composite (Supplemental Fig S4). The major exception, again, is the equatorial Pacific where we have already noted the mismatch in placement of the interannual variability in Chl-a. The model is biased low in most other regions. However, the model does capture the lower ratio of interannual variability relative to the mean in the 490 and 510nm than the other bands (Fig 3, Supplemental Fig S4)."

5. Here's an easy one... On line 288, I think 'relative' should be 'relatively' :)

Thank you, yes – now changed.

I hope these comments are helpful. Again, nice job!

Yes, these have been very useful comments, and we thank the reviewer very much.

REVIEWERS' COMMENTS:

Reviewer#3

I would like to thank the authors for their thorough response to my previous comments. This is a very well written and interesting manuscript. I have no further comments of significance. However, below is a short list of EXTREMELY minor comments the authors might want to fix before publication. Well done!

1.line 126: you can remove the comma after "growth rates"

2.line 296: you can also remove the comma here after "high latitudes"

3.line 467: "attempts" should be "attempt"

4.line 468: you can remove the word "also"

5.line 477: add a comma after "instance" and remove the comma after "light"

6.line 485-486: you can remove the comma after "discontinuities" and should add commas after "observations" and "Fig S1-S4)"

7.line 511: you can remove the comma after "real ocean"

8.line 516: you could revise this to "...have high cloud cover and, in such regions, the satellite..."

We thank the reviewer for taking the time to carefully re-read our manuscript. All changes (which are grammatical or punctuation in nature) as suggested by the reviewer have been corrected in the revised manuscript.

REVIEWERS' COMMENTS:

Reviewer#3

I would like to thank the authors for their thorough response to my previous comments. This is a very well written and interesting manuscript. I have no further comments of significance. However, below is a short list of EXTREMELY minor comments the authors might want to fix before publication. Well done!

Thank you

1.line 126: you can remove the comma after "growth rates"

Done

2.line 296: you can also remove the comma here after "high latitudes"

Done

3.line 467: "attempts" should be "attempt"

Corrected.

4.line 468: you can remove the word "also"

Agreed, now done.

5.line 477: add a comma after "instance" and remove the comma after "light"

Much better, thank you.

6.line 485-486: you can remove the comma after "discontinuities" and should add commas after "observations" and "Fig S1-S4)"

Yes, now done.

7.line 511: you can remove the comma after "real ocean"

Yes, done.

8.line 516: you could revise this to "...have high cloud cover and, in such regions, the satellite..."

This reads much better. Thank you.